# Actomyosin forces trigger a conformational change in desmoplakin within desmosomes

Yinchen Dong [1], Ahmed Elgerbi[2], Bin Xie[3], Yerin Han[4], Adam V. Kwiatkowski [4], John S. Choy [2] & Sanjeevi Sivasankar [1,3] ✉

Desmosomes are essential cell-cell adhesion organelles that enable tension-prone tissues, like the skin and heart, to withstand mechanical stress. Desmosomal anomalies are associated with numerous epidermal disorders, cardiomyopathies, and cancer. Despite their critical importance, how desmosomes sense and respond to mechanical stimuli is not understood. Here, we combine super-resolution imaging in epithelial cells and primary cardiomyocytes, FRET-based tension sensors, atomistic computer simulations, and biochemical assays to demonstrate that actomyosin forces induce a conformational change in desmoplakin, a key cytoplasmic desmosomal protein. We show that in human breast cancer MCF7 cells, keratin-19 couples F-actin filaments to desmosomes and regulates the level of actomyosin forces integrated into the desmosomal complex. We demonstrate that actomyosin contractility reorients keratin intermediate filaments and directs force to desmoplakin along the keratin network, plausibly converting the N-terminal plakin domain from a folded to an extended conformation. We also show that desmoplakin undergoes a similar actomyosin force-dependent conformational change in primary cardiomyocytes, with the extent of the change affected by myofibril orientation. Our findings establish that desmoplakin is mechanosensitive and its structural states reflect the level of forces transmitted through the actin network across cell types.

Desmosomes are critical cell-cell junctions that promote adhesion between cells and preserve tissue integrity. They enable tension-prone tissues like the skin and heart to withstand mechanical stress[1]. Desmosomal anomalies are associated with numerous diseases such as epidermal autoimmune disorders, arrhythmogenic ventricular cardiomyopathy, and cancer[2–7]. Despite their critical role in maintaining tissue mechanical resilience, an understanding of how desmosomes respond to mechanical stimuli is lacking.

Desmosomes are composed of transmembrane desmosomal cadherins, desmoglein (Dsg) and desmocollin (Dsc), and cytoplasmic plaque proteins: plakophilin, plakoglobin, and desmoplakin (DP). The extracellular regions of the cadherins from opposing cells interact to mediate adhesion, while the plaque proteins bind to the cadherin cytoplasmic tail (Fig. 1A)[8]. DP anchors desmosomes to the keratin intermediate filament (KIF) cytoskeleton, and this linkage is essential for maintaining tissue integrity and facilitating intracellular signal transduction[9,10]. Several studies show that KIFs work in cooperation with the actomyosin cytoskeleton to modulate the mechanical state of desmosomes[11–13]. Disassociating KIFs from circumferential actin belts alters tension across desmosomes[11], while disassembling the actin cytoskeleton decreases desmosome resilience[12]. However, the molecular mechanisms by which actomyosin forces alter desmosome structure and function are not understood.

[1]Department of Biomedical Engineering, University of California, Davis, CA, USA. [2]Department of Biology, The Catholic University of America, Washington, DC, USA. [3]Biophysics Graduate Group, University of California, Davis, CA, USA. [4]Department of Cell Biology, University of Pittsburgh School of Medicine, Pittsburgh, PA, USA. ✉e-mail: ssivasankar@ucdavis.edu

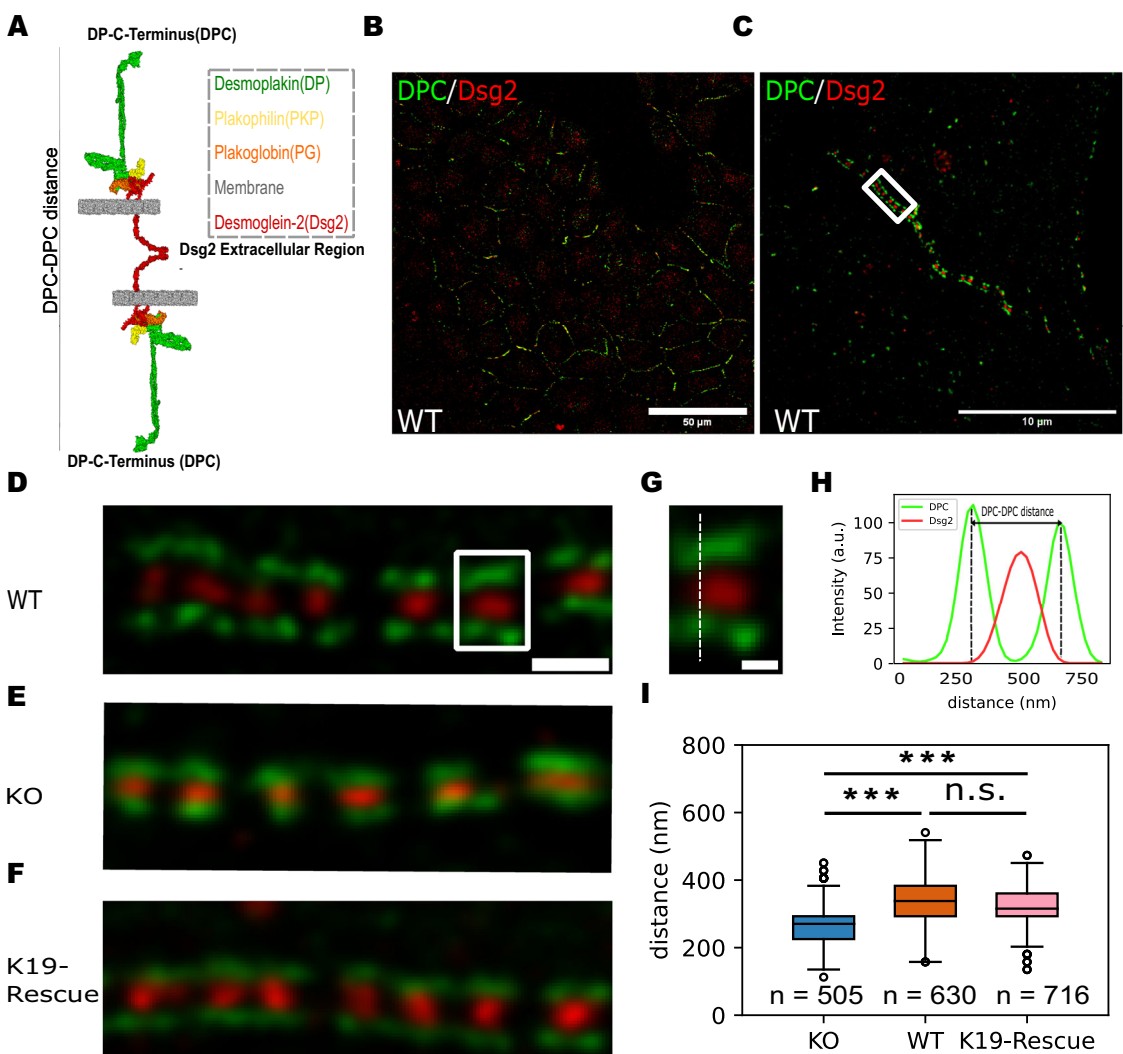

**Fig. 1 | Keratin-19 influences desmosome architecture. A** Schematic of desmosome structure. Extracellular regions of desmoglein-2 (Dsg2) interact to mediate adhesion, while their cytoplasmic tails bind to the plaque protein, plakoglobin (PG), plakophilin (PKP), and desmoplakin (DP). **B** Representative confocal image of desmosome puncta along cell-cell borders in WT cells. The desmosomes were immunolabeled for DPC (green) and Dsg2 (red). Scale bar: 50 μm. Images were taken across three biological replicates. **C** Representative STED image of desmosomes showing a characteristic 'railroad track' pattern. Scale bar: 10 μm. Images were taken across three biological replicates. Representative close-up STED images of desmosomes exhibiting the signature 'railroad track' pattern in **D** MCF7 WT cells, **E** MCF7 K19-KO cells, and **F** MCF7 K19-GFP cells. Scale bar: 500 nm. **G** The boxed area from the representative STED images of WT cells shows a closer look at an individual DP railroad track with Dsg2 between the two parallel DP plaques. Scale bar: 200 nm. **H** Line-scan analysis of DPC and Dsg2 fluorescence intensity (indicated by the dashed line in **G**). **I** Quantification of DPC-DPC distance from WT, K19-GFP, and K19-KO cells shows that desmosomes in WT and K19-GFP cells are wider than in K19-KO cells. Number of data points ($n$) = 630 (WT), 505 (K19-KO), 716 (K19-GFP); Number of replicates ($N$) = 3. Group differences were assessed with a two-sided Kruskal–Wallis test. Post-hoc pairwise comparisons used Dunn's test with Holm adjustment for multiple comparisons; adjusted $P = 1.03E-12$ (WT vs. KO), $P = 1.55E-09$ (K19-Rescue vs. KO), $P = 0.158$ (WT vs. K19-Rescue); ***$P < 0.001$; ns, $P > 0.05$. Boxplots show median, 25th and 75th percentile with whiskers reaching the last data point within 1.5× interquartile range. Data points outside this range are plotted individually as outliers. The number of data points n represents the number of line scans across the desmosomes.

Disruption of the linkage between DP and KIF impairs intercellular adhesion and compromises tissue integrity[13,14]. Knocking out keratin leads to elevated phosphorylation of DP, resulting in accelerated endocytosis of desmosomes and rupture of epithelial sheets under rotational stress[15]. Weakening connections between DP and KIF networks dramatically alters the mechanical properties of cells, such as cell stiffness and intercellular forces[12]. While these data suggest that DP is crucial in regulating the response of desmosomes to mechanical stress, the biophysical mechanism by which DP mechanically regulates the desmosome is unknown.

The structure of DP offers hints into its mechanoregulatory function. DP is a large, 2871 amino acid protein composed of an N-terminal region, a central rod domain that facilitates DP homodimerization[3], and a C-terminal region that is crucial for aligning and binding KIFs[16]. The N-terminal region of DP is subdivided into a globular head domain (the extreme N-terminal ~200 residues) and a plakin domain, each playing distinct roles[17]. The globular head domain binds to the C-termini of desmosomal cadherins and recruits plakophilin and plakoglobin[18]. The plakin domain, characterized by six spectrin repeat domains interspersed with Src homology domains, exhibits structural flexibility, as observed in a previous in vitro, cell-free small-angle X-ray scattering study[19]. That investigation revealed that the plakin domain adopts either a folded "U" shape or a fully extended "I" shape, indicating a significant degree of structural adaptability[20]. This finding suggests that actomyosin forces could induce a conformational change in the N-terminal region of DP,

thereby offering a molecular pathway for DP mechanosensation. However, force-induced conformational changes in DP, particularly within the cell, have never been demonstrated.

DP exists in two major splice isoforms, I and II, which differ only in the length of their rod domain. Our study focuses on DP isoform I (DPI), which is expressed at high levels in epithelia and cardiomyocytes[21]. We demonstrate, in both human breast cancer MCF7 cells and primary cardiomyocytes, that actomyosin forces induce a conformational change in DPI. Since MCF7 cells are highly enriched in keratin 19 (K19), we interrogated wild-type (WT), K19-knockout (K19-KO), and K19-GFP rescue (K19-GFP) cells using confocal and super-resolution microscopy, FRET-based tension sensors, pharmacological inhibition and activation of actomyosin contractility, and co-immunoprecipitation (co-IP). We show that K19 reorganizes actin and regulates the level of actomyosin forces integrated into the desmosomal complex. We demonstrate that actomyosin contractility reorients the KIF network and directs force to DPI along K19 filaments, thereby altering DPI conformation. Using super-resolution microscopy, we demonstrate that actomyosin forces similarly change DPI conformation in mechanically active primary cardiomyocytes. Similar to MCF7 cells, where DPI conformational change depends on actin filament orientation, we show that DPI conformational change in cardiomyocytes depends on myofibril orientation. Atomistic Molecular Dynamics (MD) simulations suggest that actomyosin forces change the conformation of the N-terminal plakin domain, from a folded (closed) to an extended (open) conformation. Our findings establish, at the molecular level, that DP is a mechanosensor within the desmosome.

## Results

### K19 remodels desmosome architecture

Desmosomes are characterized by distinct "railroad" track structures, represented by discrete parallel fragments of DP, which sandwich the desmosomal cadherins[22]. To validate this signature structure of desmosomes in MCF7 cells, we conducted immunostaining targeting the extracellular region of isoform 2 of desmoglein (Dsg2) and the C-terminal region of DPI (DPC) using confocal and super-resolution Stimulated Emission Depletion (STED) microscopy. Confocal imaging revealed colocalization of DPC and Dsg2 signals, indicating the formation of desmosome puncta along cell–cell junctions (Fig. 1B). STED imaging in WT cells showed well-defined DP "railroad" tracks, as depicted in Fig. 1C, D. To investigate the effect of K19 on desmosome architecture, we used the K19-KO cell line[23]. Successful ablation of K19 in these cells was confirmed using western blotting (Fig. S1). Immunoblotting also showed that knocking out K19 did not significantly alter the levels of desmosomal cadherins (Dsg2, Dsc1, and Dsc2-3), plaque proteins (plakophilin-1, plakophilin-3, and DP), or cytoskeletal proteins, including K8 and K18, F-actin, and myosin heavy chain 10 (Fig. S1). Surprisingly, in the K19-KO cells, STED imaging revealed more compact desmosomes (Fig. 1E). To quantify desmosome width, we measured the distance between opposing DP plaques in the STED images by drawing a line scan across the desmosome complex (Fig. 1G) and measuring the distance between the two peaks of DPC signal (designated as DPC-DPC distance) in the fluorescence intensity plot (Fig. 1H). This quantitative analysis confirmed that desmosomes in K19-KO cells were significantly narrower than desmosomes in WT MCF7 cells ($335 \pm 61$ nm for WT cells and $264 \pm 65$ nm for K19-KO cells; Fig. 1I).

To corroborate that the observed shift in desmosome width was directly related to the absence of K19, we stably rescued the K19-KO cells with EGFP-tagged K19 (K19-GFP cells). Western blots validated the successful expression of K19-GFP in the rescued cells (Fig. S2). STED imaging showed that desmosome width in K19-GFP cells recovered to a level similar to WT cells ($321 \pm 57$ nm; Fig. 1F, I). Together, these findings demonstrate that desmosome architecture in MCF7 cells is strongly correlated with the presence/absence of K19.

### The interplay between K19 and the actomyosin network governs desmosome width

The cytoplasmic KIF network binds to DP at its C-terminus (illustrated in Fig. 2A), and interactions between the KIF network and elements of the actomyosin cytoskeleton have been observed at cell-cell junctions and the cell-substrate interface[11,24]. To investigate how changes in the KIF network contribute to desmosome remodeling, we first inspected the intracellular localization of K19, DP, and F-actin filaments in K19-GFP cells. K19 was found to be colocalized with the DP "railroad" tracks along the cell border (Fig. 2B). Additionally, actin filaments were observed to align with the K19 filaments (Fig. 2B). Based on the colocalization of K19 with DP and the coalignment of K19 with F-actin filaments, we hypothesized that K19 interacts with both DP and F-actin filaments.

To validate this hypothesis, we conducted co-IP experiments using IgG as the control and K19 as the bait protein. The results revealed that K19 interacts with both DP and actin, consistent with our previous immunofluorescence microscopy observations (Fig. 2C). To confirm that the proteins pulled down with K19 were due to specific interactions, we performed a K19 pulldown in K19-KO cells. Neither DP nor actin filaments were detected in the pulldown from K19-KO cells (Fig. S3), supporting that their association with K19 in WT cells reflects specific interactions. To further confirm the specificity of K19 interactions with DP and actin, we blotted for the membrane-associated protein E-cadherin in K19 pulldowns from WT and K19-KO cells. E-cadherin was not detected in the pull-downs, indicating that K19 associations with DP and actin were specific and not due to general membrane localization (Fig. S4). Additionally, given the high abundance of keratin and actin in cells and their potential for indirect crosstalk, we performed a DP pulldown in both MCF7 WT and K19-KO cells. These experiments revealed that DP interacts with keratin and actin filaments in both cell types, supporting a specific linkage between DP and the cytoskeletal networks (Fig. S5). Finally, the Co-IP experiments showed that in addition to DP, the desmosomal plaque protein plakophilin-3 also interacted with K19 (Fig. S6), which further supports the interaction between K19 filaments and the desmosome complex. While our Co-IP results confirmed the interactions between K19 and desmosomal proteins, we note that these interactions may occur outside of desmosomes.

It has previously been demonstrated that actin stress fiber formation strongly correlates with the presence of keratin filaments in cells[25]. We therefore used confocal fluorescence microscopy to determine if actin stress fibers, which are a proxy for intracellular forces, are present in WT, K19-GFP, and K19-KO cells (Fig. S7A). Quantitative data analysis showed that, compared to WT and K19-GFP cells, actin stress fibers in K19-KO cells were dramatically decreased (Fig. S7B). It is worth noting that while the formation of actin stress fibers reflects mechanical tension from actomyosin contractility, the reduction in stress fibers in K19-KO cells may also be influenced by broader changes in cytoskeletal regulation beyond a direct loss of tension. Nonetheless, the observed phenotype prompted us to hypothesize that the KIF network may facilitate the transmission of actomyosin-generated forces to desmosomes, potentially acting as a regulatory bridge between the actin network and the desmosome complex.

To test this hypothesis, we first used Blebbistatin, a myosin-II-specific ATPase inhibitor, to reduce the actomyosin contractility in the WT cell line (WT+Blebb) and measured the desmosome width. STED imaging showed more compact desmosomes in the WT+Blebb cells (Fig. 2D); desmosome width in WT+Blebb cells was reduced to a similar level as K19-KO cells ($263 \pm 62$ for WT+Blebb vs. $264 \pm 65$ nm for K19-KO; Fig. 2E). At the same time, the control group treated with dimethyl sulfoxide (DMSO, Blebbistatin's solvent), retained similar desmosome widths as the WT cells (Fig. S8A).

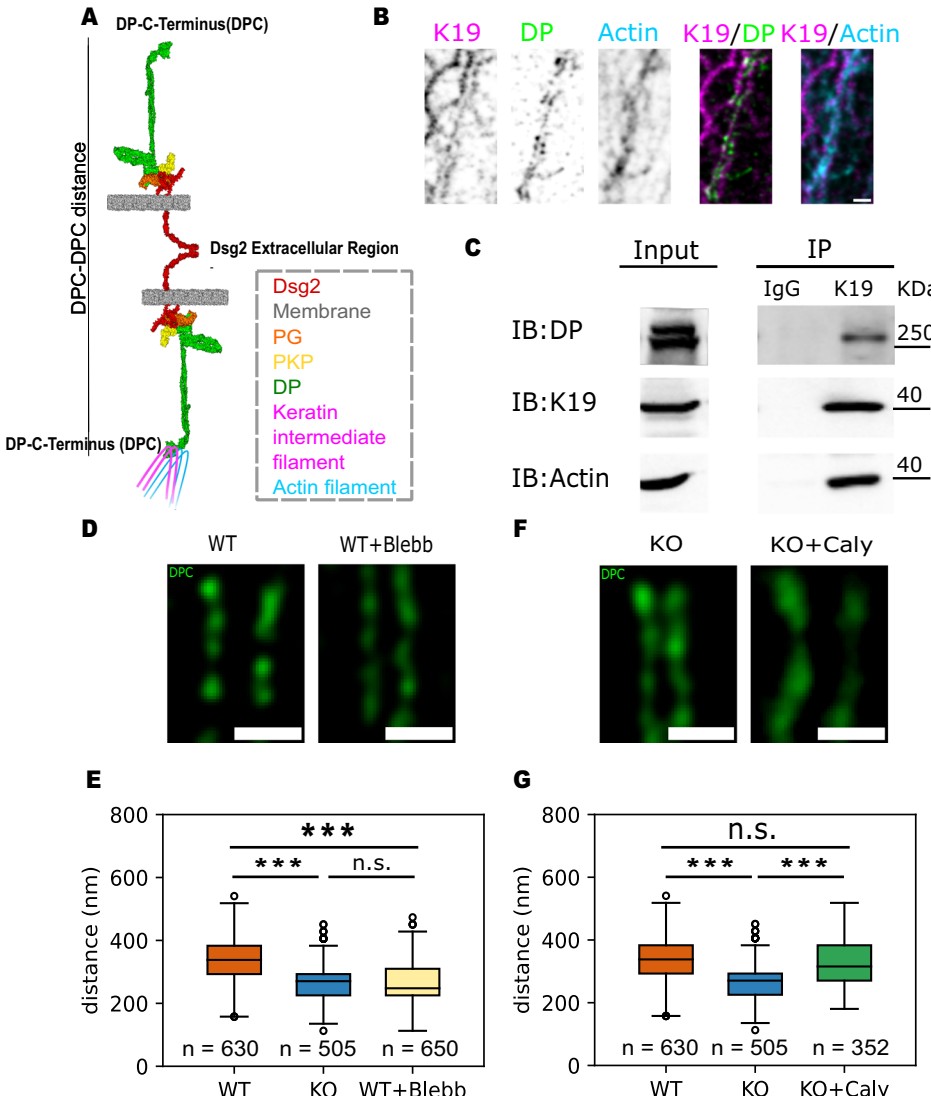

**Fig. 2 | Interplay between keratin-19 and actomyosin network determines desmosome width. A** Schematic of the interactions between the desmosome keratin intermediate filaments, and actin filaments. Keratin filaments bind to the C-terminal of DP and interact with actin filaments in the cytoplasm. **B** Representative STED images of K19-GFP rescue cells immunolabeled for K19 (magenta), DP (green), and actin (cyan). The individual components are shown in grayscale to highlight the structural features. The merged images between K19 and DP or actin are shown at the right. Scale bar: 500 nm. The images show that K19 filaments colocalize with the DP "railroad" tracks and with cortical actin belts. Images were taken across three biological replicates. **C** Co-IP in WT cells performed with anti-K19 antibody or IgG control. The co-IPs show that K19 interacts with both DP and actin. Experiments were done across three biological replicates. **D** Representative STED images of DPC (green) at the cell border of WT cells and Blebbistatin-treated WT cells (WT+Blebb). **E** Quantification of desmosome widths from WT, K19-KO, and WT+Blebb cells showing that Blebbistatin treatment reduces the desmosome width to a similar level as in the K19-KO cell line. $n = 630$ (WT), 505

(K19-KO), 650 (WT+Blebb); $N = 3$. Group differences were assessed with a two-sided Kruskal–Wallis test. Post-hoc pairwise comparisons used Dunn's test with Holm adjustment for multiple comparisons; adjusted $P = 7.37E{-}11$ (WT vs. KO), $P = 1.05E{-}12$ (WT+Blebb vs. WT), $P = 0.847$ (WT+Blebb vs. KO); ***$P < 0.001$; ns, $P > 0.05$. **F** Representative STED images of DPC (green) at the cell border of K19-KO cells and Calyculin A-treated K19-KO cells (KO+Caly). **G** Quantification of desmosome widths from WT, K19-KO, and KO+Caly cells showing that Calyculin A increases the desmosome width of K19-KO cells to a similar level as in the WT cell line. $n = 630$ (WT), 505 (K19-KO), 353 (KO+Caly); $N = 3$. Group differences were assessed with a two-sided Kruskal–Wallis test. Post-hoc pairwise comparisons used Dunn's test with Holm adjustment for multiple comparisons; adjusted $P = 1.08E{-}07$ (WT vs. KO), $P = 2.01E{-}05$ (KO+Caly vs. KO), $P = 0.736$ (WT vs. KO+Caly); ***$P < 0.001$; ns, $P > 0.05$. All boxplots show median, 25th and 75th percentile with whiskers reaching the last data point within 1.5× interquartile range. Data points outside this range are plotted individually as outliers. The number of data points n represents the number of line scans across the desmosomes.

Next, to test if stimulating actomyosin contractility increases desmosome width, we treated the K19-KO cells with Calyculin A, a phosphatase inhibitor (KO+Caly). STED imaging showed that the width of desmosomes in the KO+Caly cells increased to a level comparable to WT cells (Fig. 2F). Quantitative analysis of the images showed that upon Calyculin A treatment, the desmosome width in K19-KO cells was $333 \pm 83$ nm (vs. $334 \pm 61$ nm for WT cells; Fig. 2G). In contrast, the control group treated with DMSO, the solvent for Calyculin A, exhibited

desmosome widths comparable to untreated K19-KO cells (Fig. S8B). Taken together, these data demonstrate that the K19-dependent changes in desmosome architecture are a consequence of a mechanical pathway established due to interactions between K19, F-actin, and DP. Consequently, contractile actomyosin forces are transmitted to and reorganize the desmosome, increasing desmosome width.

While our data demonstrates that K19 is critical in transmitting actomyosin force to desmosomes, the molecular mechanism

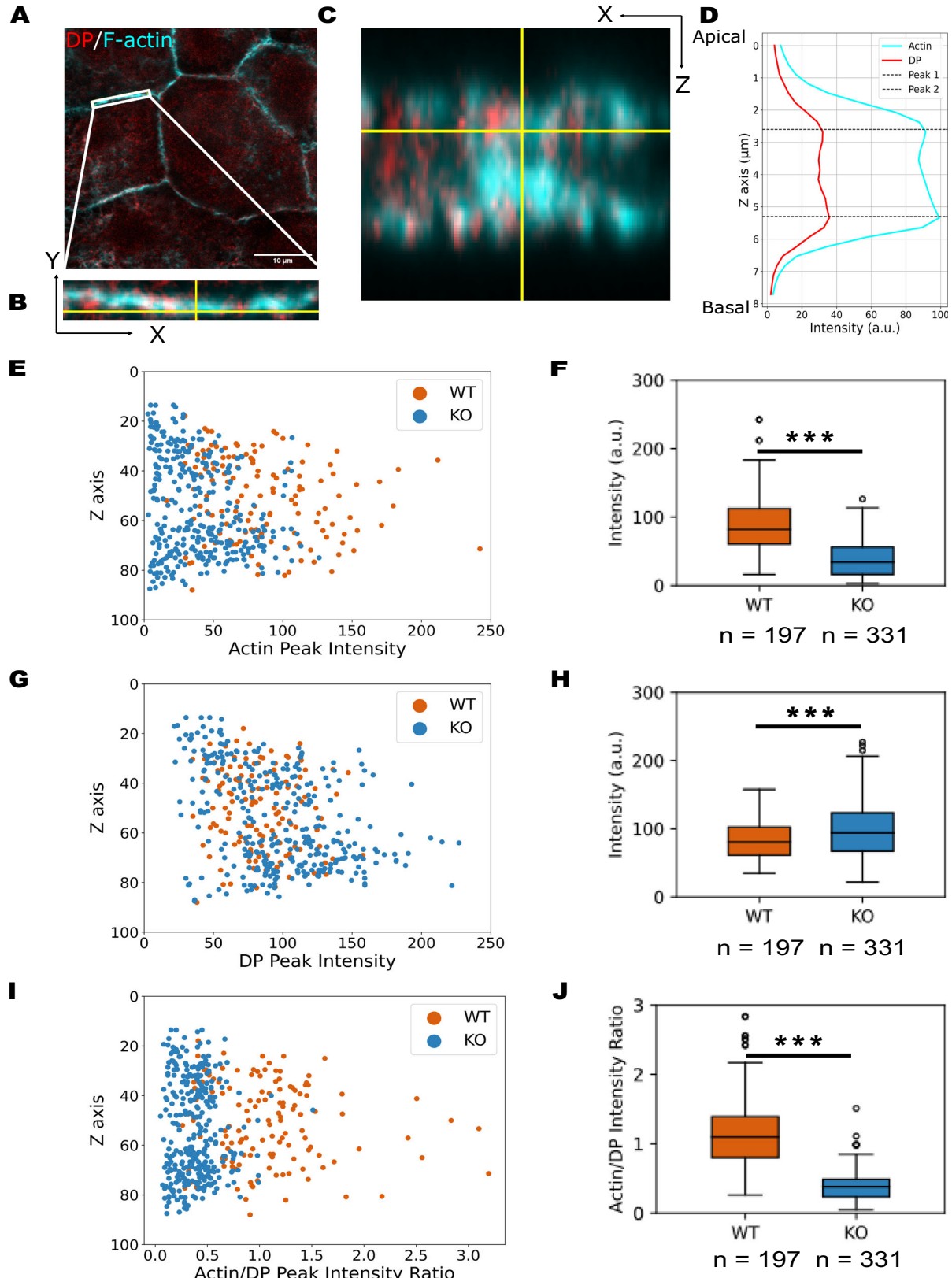

underlying K19's crucial role remains unresolved. Previous studies in Caco-2 cells have shown that K19 knockdown significantly reduces apical F-actin filaments, suggesting a role for K19 in organizing the actin cytoskeleton[26]. This lead us to hypothesize that K19 may play a similar role in MCF7 cells–helping to organize F-actin filaments and thereby facilitating the transmission of actomyosin-generated forces

to desmosomes. To investigate whether K19 regulates the organization of F-actin and its coupling to desmosomes, we acquired Z-stack confocal images of MCF7 WT and K19-KO cells stained for DP and F-actin. For each image stack, we identified the XY plane with the highest DP signal (Fig. 3A) and selected a region of interest (ROI) based on this projection (Fig. 3B). To analyze signal distribution along the Z-axis, we

**Fig. 3 | KIF network regulates F-actin coupling to desmosomes.**
**A** Representative confocal overlay image of XY maximum intensity projection of DP (red) and F-actin (cyan) in MCF7 cells. Scale bar: 10 μm. Images were taken across three biological replicates. **B** Magnified view of the boxed area in (**A**) illustrates the selected ROI at a specific cell-cell junction. **C** XZ plane of cross-sectional view of the ROI demonstrates the distribution of DP (red) and F-actin (cyan) signals along the X- and Z-axes. **D** Fluorescence intensity profile of DP and F-actin reveals peak intensities and corresponding Z positions at the junction. **E** Scatter plot of actin peak intensities along the Z-axis in WT and K19-KO cells shows generally higher actin peak intensities in WT compared to K19-KO cells. **F** Quantification of actin peak intensities along the Z-axis from WT and K19-KO cells. $n = 197$ (WT), 331 (K19-KO); $N = 3$. Two-sided Mann–Whitney's U test; $P = 4.287E\text{-}44$, ***$P < 0.001$. Actin peak intensities along the junction are significantly higher in the WT compared to the K19-KO cells. **G** A scatter plot of DP peak intensities along the Z-axis in WT and K19-KO cells shows overlapping distributions between WT and K19-KO cells. **H** Quantification of DP peak intensities along the Z-axis from WT and K19-KO cells. $n = 197$ (WT), 331 (K19-KO); $N = 3$. Two-sided Mann–Whitney's U test; $P = 1.32E\text{-}4$, ***$P < 0.001$. The DP peak intensities along the junction are slightly greater in the K19-KO compared to the WT cells. **I** A scatter plot of actin/DP peak intensity ratios along the Z-axis from WT and K19-KO cells shows a clear separation between the WT and K19-KO cells. **J** Quantification of actin/DP peak intensity ratios along the Z-axis from WT and K19-KO cells. $n = 197$ (WT), 331 (K19-KO); $N = 3$. Two-sided Mann–Whitney's U test; $P = 8.435E\text{-}64$, ***$P < 0.001$. The actin/DP peak intensity ratio along the junction is significantly greater in the WT compared to the K19-KO cells. All boxplots show median, 25th and 75th percentile with whiskers reaching the last data point within 1.5× interquartile range. Data points outside this range are plotted individually as outliers. The number of data points n represents the number of peak intensities for the corresponding Z-axes.

generated an XZ cross-section by slicing along the X-axis through the defined ROI (Fig. 3C). We quantified the distribution of DP and F-actin by measuring their peak intensities and corresponding Z positions from fluorescence intensity profiles (Fig. 3D). Notably, each DP peak was accompanied by an adjacent F-actin peak at a similar Z position, consistent with our model in which F-actin filaments associate with desmosomes through KIF network.

To examine how K19 influences the organization of the actin network, we plotted the F-actin peak intensity as a function of positioning along the Z-axis for both WT and K19-KO MCF7 cells (Fig. 3E). Notably, WT cells exhibited significantly higher F-actin peak intensity compared to K19-KO cells throughout the Z-axis (90.3 ± 41.5 a.u. in WT vs. 38.9 ± 27.0 a.u. in K19-KO; Fig. 3F), consistent with prior reports of reduced apical actin filaments upon K19 depletion. In contrast, DP peak intensity was comparable between the two groups (Fig. 3G), with K19-KO cells showing slightly higher values (82.9 ± 26.6 a.u. in WT vs. 96.7 ± 40.7 a.u. in K19-KO; Fig. 3H). To further explore how K19 may regulate F-actin coupling to desmosomes, we calculated the ratio of F-actin to DP peak intensity across the Z-axis. This ratio showed a clear separation between WT and K19-KO cells (Fig. 3I), with WT cells displaying a roughly threefold higher actin/DP ratio (1.16 ± 0.56 in WT vs. 0.38 ± 0.20 in K19-KO; Fig. 3J). Taken together, these results indicate that K19 plays a key role in organizing F-actin filaments and anchoring them to desmosomes. In the absence of K19, the F-actin filament association with desmosomes is markedly reduced, which impairs the transmission of actomyosin-generated forces to these adhesive sites.

## DP undergoes a conformational change when pulled

Since previous studies have reported structural flexibility in the N-terminal plakin domain of DP[20], we hypothesized that the increase in desmosome width in WT cells was due to a force-induced conformational change in this region of DP (Fig. 4A). To test this hypothesis, we specifically stained both the DPI N-terminal region (DPN) on its globular head domain (which is upstream of the plakin domain) and the extracellular region of Dsg2. Super-resolution imaging revealed wider desmosomes in WT cells stained with DPC and Dsg2 compared to K19-KO cells (Fig. 4B). In contrast, both cell lines exhibited similar desmosome widths when they were stained with DPN and Dsg2 (Fig. 4C). To gain a clearer understanding of the DP structural changes in WT and K19-KO cell lines, we measured the distances between Dsg2 and DPC, as well as Dsg2 and DPN, within the half-unit of the desmosome complex. This analysis showed that while Dsg2-DPC distance was significantly greater in WT compared to the K19-KO cells (168 ± 47 nm for WT cells and 132 ± 49 nm for K19-KO cells; Fig. 4D), Dsg2-DPN distance was similar in both cell lines (99 ± 44 nm for WT cells and 101 ± 53 nm for K19-KO cells; Fig. 4E). As seen in Fig. 4F, the DP length (defined as the distance between DPN and DPC) was 33 nm greater in the WT cells compared to the K19-KO cells, suggesting that DP was elongated by ~33 nm in the WT cell lines due to higher mechanical tension in these cells.

To explore the potential structural mechanism underlying the observed DP extension in response to forces, we proceeded to atomistically simulate the effect of force on the conformation of the highly conserved plakin domain of DP (residues 178-950). We first generated the entire plakin domain structure using AlphaFold2[27]. The predicted structure exhibited a 'U-shaped' folded conformation, with spectrin repeat (SR) domains SR3-6 and Src homology (SH) domain SH3 forming a long arm that interacts with a short arm composed of SR7-8 and SR8-CT domains (Fig. 4G). This prediction agrees well with the crystal structure of the long arm of the plakin domain[28] (RMSD - 1.7Å, Fig. S9).

To identify the interactions crucial for stabilizing the folded conformation, we performed three force-independent molecular dynamics (MD) simulations on the predicted structure. In the MD simulations, the plakin domain stabilized within 20 ns (Fig. S10). During the simulations, 10–15 hydrogen bonds were observed between the long arm and the short arm, with interactions primarily occurring between the SR5-6 and SR7-8 regions (Fig. 4H and Fig. S11). Notably, two consistent salt bridge/hydrogen bond interactions, between four conserved amino acids, were observed across all three simulations: R592-N744 and K444-E858 (Fig. 4G).

To test if the folded plakin domain structure can be unfolded using pulling forces, we performed steered molecular dynamics (SMD) simulations (Supplementary Movie 1). In the SMD simulations, we fixed the position of the N-terminal residues and pulled the C-terminal regions (residues 919-950) using a constant force. Our simulations showed that the plakin domain's long and short arms separated due to the pulling force without any secondary structure unfolding (Fig. 4J). Consequently, the plakin domain transitioned from the initial closed (Fig. 4J, left) to the final open (Fig. 4J, right) conformation (Supplementary Movie 1). In the open conformation, the distance between the plakin domain's C-terminus and N-terminus increased by 33 nm compared to the closed structure (Fig. 4I), which is consistent with the change in DPC-DPC distance observed in the STED images (Fig. 4F). The structural predictions and SMD simulations suggest that DP adopts a U-shaped conformation under low tension and an extended conformation under load.

## Keratin intermediate filaments reorient in response to changes in actomyosin forces

Given that actomyosin forces are transmitted to desmosomes via KIFs, a key question is how this force transmission is affected by the organization of keratin filaments. A previous study in *Xenopus laevis* showed that the KIF network reorganizes in response to force transmission across cell-cell contacts and becomes aligned with the direction of protrusive activity and tissue movement[29]. This raises the question as to whether a similar phenomenon occurs in MCF7 cells (Fig. 5A).

In epithelial cells, KIFs are composed of heterodimers between type I and type II keratins, which assemble into antiparallel

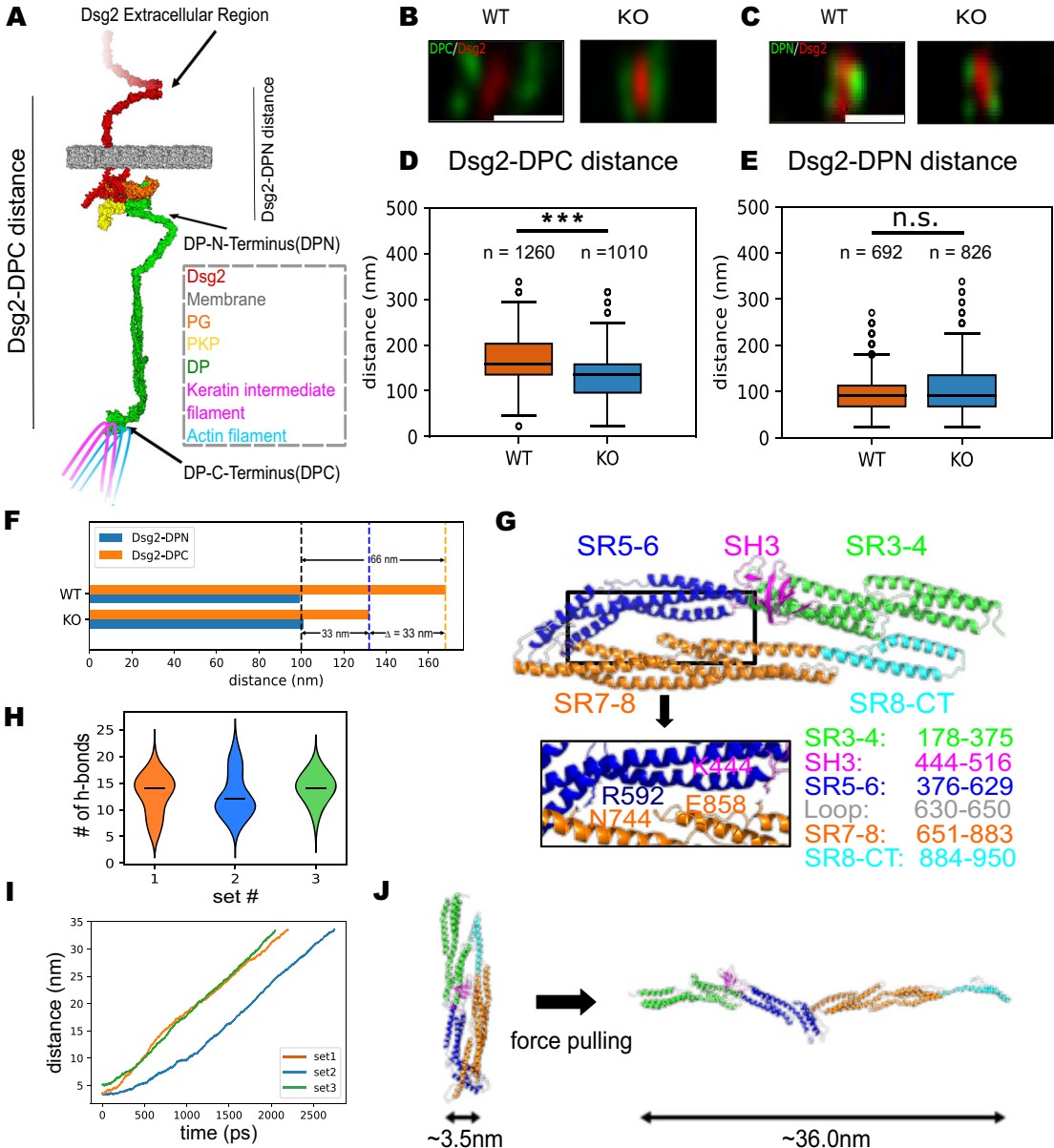

**Fig. 4 | Desmoplakin undergoes force-induced conformational change in the plakin domain. A** Schematic of the desmosome under tension. The N-terminal region of DP transitions from a closed conformation to an open conformation. **B** Representative STED images of DPC (green) and Dsg2 (red) at the cell border for WT and K19-KO cells. Scale bar: 500 nm. **C** Representative STED images of DPN (green) and Dsg2 (red) at the cell border for WT and K19-KO cells. Scale bar: 500 nm. **D** Quantification of desmosome half-unit widths (Dsg2-DPC distance) from WT and K19-KO cells. $n = 1260$ (WT), 1010 (K19-KO); $N = 3$. Two-sided Mann–Whitney's U test; $P = 1.542E{-}66$, ***$P < 0.001$. The distances between Dsg2 and DPC are significantly greater in the WT compared to the K19-KO cells. **E** Quantification of desmosome half-unit widths (Dsg2-DPN distance) from WT and K19-KO cells. $n = 692$ (WT), 826 (K19-KO); $N = 3$. Two-sided Mann–Whitney's U test; ns, $P > 0.05$. The Dsg2-DPN distance is similar in both cell lines. **F** The mean values from the results in D and E are summarized in the bar chart to compare the DP length (DPN - DPC distance) between the WT and K19-KO cells. The DP length is 66 nm for the WT and 33 nm for the K19-KO, indicating that DP extends 33 nm in

the WT cells. **G** The entire plakin domain structure predicted by AlphaFold2 suggests a U-shape conformation in the absence of force. Two common electrostatic interactions (R592-N744 and K444-E858) observed in all 3 MD simulations (below) are shown. **H** During the MD simulations, 10 to 15 hydrogen bonds were formed between the plakin domain's long arm (SR3-4 and SR5-6) and short arm (SR7-8 and SR8-CT). **I** During the SMD simulations, the distance between the plakin domain's C-terminus and N-terminus increased by 30-33 nm. This elongation correlates with the distance change experimentally measured in Fig. 3F. **J** Comparison of the plakin domain structure at the start (left) and at the end (right) of the SMD simulation. The structures show that the plakin domain was elongated by the pulling force without significant secondary structure unfolding. All boxplots show median, 25th and 75th percentile with whiskers reaching the last data point within 1.5× interquartile range. Data points outside this range are plotted individually as outliers. The number of data points n represents the number of line scans across the desmosomes.

tetramers[30]. A previous RNA-sequencing study revealed that type II Keratin-8 (K8) was the most highly expressed isoform in the MCF7 cells, followed by roughly equal amounts of type I Keratin-19 (K19) and type I Keratin-18 (K18)[31]. This suggests that the primary keratin pairs in MCF7 cells are K8/K19 and K8/K18. Notably, K8 and K18

expression levels remained unchanged when K19 was knocked out (Fig. S1), which allowed us to stain K8 and K18 to investigate KIF organization. When we performed Co-IP experiments using K8 and K18 as bait, we observed that both keratin isoforms interacted with DP and actin filaments in WT and K19-KO cells (Fig. 5B). This

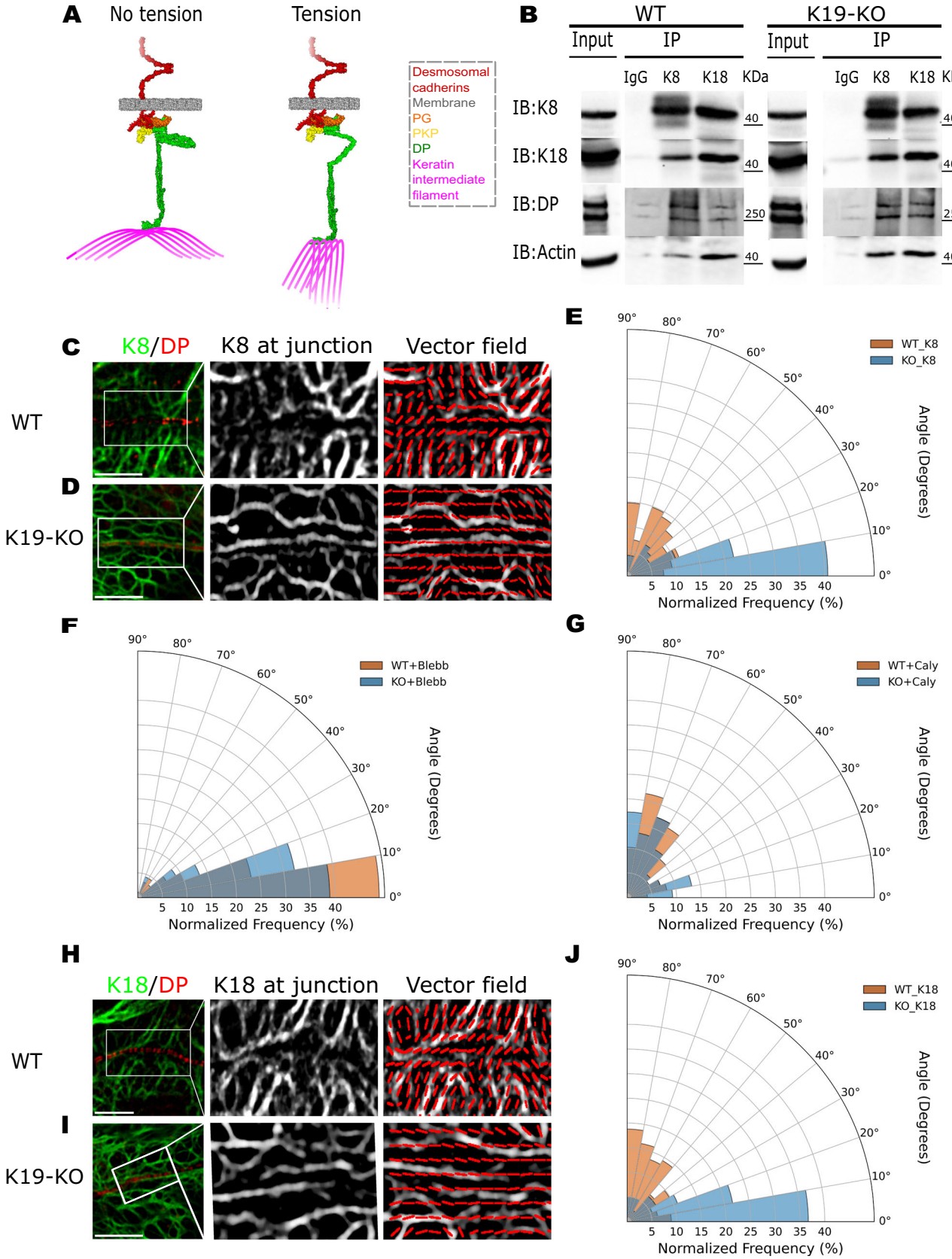

suggested that K8/K18 filaments could also transmit actomyosin forces to desmosomes.

We first imaged the organization of K8 filaments in WT and K19-KO cells by staining both K8 and DP. In WT cells, the K8 filaments at junctions exhibited a more radial organization (Fig. 5C), which exposes

the desmosome to normal forces that can presumably stretch DP. In contrast, the K8 filaments in K19-KO cells were primarily arranged horizontally (Fig. 5D), almost parallel to the desmosome, which presumably results in tangentially actomyosin forces that cannot stretch DP. We quantified filament organization by determining the local

**Fig. 5 | Keratin filaments reorient in response to changes in actomyosin forces.**
**A** Schematic representation of DP-associated keratin filaments at the cell-cell junction. Under no-tension conditions, DP-associated filaments are primarily aligned horizontally. Under tension, they transition to a radially organized structure that exerts force on the desmosome. **B** Co-IP of K8 and K18 in WT and K19-KO cells. Co-IP was performed using anti-K8, anti-K18 antibodies, or an IgG control. The co-IPs reveal that both K8 and K18 interact with DP and actin. Experiments were done across three biological replicates. Representative confocal images of WT (**C**) and K19-KO (**D**) cells immunolabeled for K8 (green) and DP (red). Scale bar: 5 μm (left panel). A magnified grayscale image highlights DP-associated K8 filaments at the junction with enhanced contrast (middle panel). The local dominant orientation of K8 filaments is represented as a vector field image (right panel). **E** Polar histogram of the global dominant orientation of K8 filaments in WT and K19-KO cells. $n$ = 108 (WT), 96 (K19-KO); $N$ = 3. Two-sided Mann–Whitney U test; $P$ = 5.885E−12, ***$P$ < 0.001. The K8 filaments in WT cells exhibit a more radial organization, while K19-KO filaments are horizontally aligned. **F** Polar histogram of K8 filament orientation in WT and K19-KO cells treated with Blebbistatin. $n$ = 90 (WT), 90 (K19-KO);

$N$ = 3. Two-sided Mann–Whitney U test; ns, $P$ > 0.05. Following Blebbistatin treatment, WT filaments shift from a radial to a horizontal organization, making their global dominant orientation indistinguishable from K19-KO cells. **G** Polar histogram of K8 filament orientation in WT and K19-KO cells treated with Calyculin A. $n$ = 98 (WT), 98 (K19-KO); $N$ = 3. Two-sided Mann–Whitney U test; ns, $P$ > 0.05. After Calyculin A treatment, K8 filaments in K19-KO cells shift from a horizontal to a radial organization, making their global dominant orientation indistinguishable from WT cells. Representative confocal images of (**H**) WT and (**I**) K19-KO cells immunolabeled for K18 (green) and DP (red). Scale bar: 5 μm (left panel). Magnified images highlighting K18 filaments at the junction and local dominant orientation of K18 filaments are shown in the middle and right panel. **J** Polar histogram of the global dominant orientation of K18 filaments in WT and K19-KO cells. $n$ = 112 (WT), 120 (K19-KO); $N$ = 3. Two-sided Mann–Whitney U test; $P$ = 1.329E-12, ***$P$ < 0.001. The global dominant orientation of K18 filaments in WT cells is significantly different from that in K19-KO cells, with WT filaments adopting a radial organization while K19-KO filaments remain horizontal, mirroring the pattern observed for K8 filaments.

---

orientation, defined as the angle of the most dominant filament direction with respect to the primary axis of the cell junction (set at 0°), and aggregated the local orientations into a polar histogram (Fig. 5C, D). Our quantitative analysis showed that over 60% of WT K8 filaments had a dominant orientation between 40° and 90°, whereas more than 60% of K19-KO K8 filaments were oriented between 0° and 20° (Fig. 5E).

When actomyosin force was inhibited via Blebbistatin treatment, the K8 filament orientation in the WT cells changed dramatically to horizontally aligned (Fig. S12A) with over 70% of filaments oriented between 0° and 20° (Fig. 5F). In contrast, K19-KO cells treated with Blebbistatin showed no significant change in K8 filament orientation (Fig. S12B & Fig. 5F). On the other hand, when actomyosin contractility in the cells was enhanced using Calyculin A treatment, K8 filament organization in K19-KO cells changed dramatically to a radial organization (Fig. S13B) with over 60% of filaments having a dominant filament orientation between 40° and 90° (Fig. 5G). In contrast, K8 filament orientation in WT cells treated with Calyculin A remained largely unchanged (Fig. S13A & Fig. 5G). At the same time, the control group treated with Blebbistatin and Calyculin A's solvent, DMSO, retained similar keratin organizations as the WT and K19-KO cells (Fig. S14).

Next, to distinguish between K8/K19 and K8/K18 filaments, we also imaged K18 filament organization in WT and K19-KO cells. As observed in our K8 filament images, staining for K18 and DP revealed more radially organized filaments in WT cells and horizontally organized filaments in K19-KO cells (Fig. 5H–J). Taken together, these findings indicated that KIFs reorganize in response to changes in actomyosin forces, shifting from predominantly 'poor force-transmitting' horizontal orientation in a low-tension condition to a more 'efficient force-transmitting' radial organization under load. Consequently, K19-KO and WT+Blebb cells exhibit horizontally oriented KIF networks and more compact desmosomes, while WT cells and KO+Caly display a radially organized KIF network and wider desmosomes.

### Desmoplakin is mechanically loaded in WT cells

Next, we proceeded to directly compare tension across DP in WT and K19-KO desmosomes using a recently published DPI Förster Resonance Energy Transfer (FRET) tension sensor[32]. This F40-based tension sensor (TS) is comprised of a tension-sensing linker peptide (GPGGA)$_8$ flanked by the donor fluorophore mTFP1 and the acceptor fluorophore mEYFP, that is inserted into an unstructured region following the rod domain, just before Pro1946 in DPI (Fig. 6A)[32]. Additionally, we used a no-tension control consisting of a truncated DP that lacked the C-terminal keratin-binding domain to account for local environmental changes that may affect FRET efficiencies (Fig. 6A). While these truncated DP controls still localized to intercellular junctions, they were

unable to interact with keratin filaments and thus could not bear forces[14,32,33]. Lastly, a donor-only control was also included to determine the FRET efficiency. We transiently expressed the DP constructs in WT and K19-KO cells and analyzed the fixed samples. DP-TS localized to cell-cell contacts, with distinct DP puncta observed in both WT and K19-KO cells (Fig. 6B). Similarly, the truncated DP control and donor-only control exhibited the expected subcellular localization at cell junctions, forming clear desmosomal puncta (Fig. S15). These results indicate that all three DP constructs were correctly localized to cell-cell contacts and successfully formed desmosomes, enabling the measurement of DP tension at the desmosome.

Next, we quantified DP-TS FRET efficiencies using fluorescence lifetime imaging microscopy (FLIM) in confluent WT and K19-KO monolayers following the workflow illustrated in Fig. 6C. When imaging each transfected cell, we first isolated individual DP puncta and measured their lifetime using a bi-exponential fit, accounting for the effects of the instrument response function (IRF) and auto-fluorescence, which yielded short lifetime measurements. Finally, FRET efficiency was determined based on these lifetime measurements. In the WT cells, the FRET efficiency was significantly lower in the TS (8.80% ± 2.76%) compared to the control (18.5% ± 4.46%), indicating that DP in these cells experienced substantial mechanical loading (Fig. 6D). In contrast, the FRET efficiency of TS (18.3% ± 3.96%) in K19-KO cells was similar to the control (19.0% ± 3.88%) (Fig. 6E), suggesting that no detectable tension was present across DP in the K19-KO cells. Based on the 1–6 pN sensitivity of the F40 tension sensor and its averaged FRET–force calibration curve[32,34], we estimated that the tension in WT cells was close to the upper range, around 5 pN. These direct measurements of forces experienced by DP revealed high tension in WT cells and no detectable tension in K19-KO cells, consistent with our previous analysis of actin stress fibers.

To further validate our mechanical model, we used FLIM-FRET to examine the role of actomyosin contractility in regulating DP tension, by treating both WT and K19-KO cells with Blebbistatin to inhibit actomyosin activity (Fig. S16). In WT cells treated with Blebbistatin (WT+Blebb), the FRET efficiency of the DP TS increased markedly to 17.0% ± 1.82%, reaching a level comparable to the control (18.5% ± 2.23%) (Fig. 6F). This suggests a significant reduction in DP tension due to decreased actomyosin contractility. In contrast, K19-KO cells treated with Blebbistatin (KO+Blebb) showed no significant change in FRET efficiency between the TS (17.9% ± 2.13%) and the control (17.7% ± 2.46%) (Fig. 6G), indicating that DP in K19-KO cells already experiences minimal mechanical tension. We were unable to perform similar experiments under calyculin A treatment due to the low transfection efficiency of the DPI-TS constructs coupled with calyculin A mediated rupture of cell junctions. These findings further support our model in which actomyosin-generated forces are

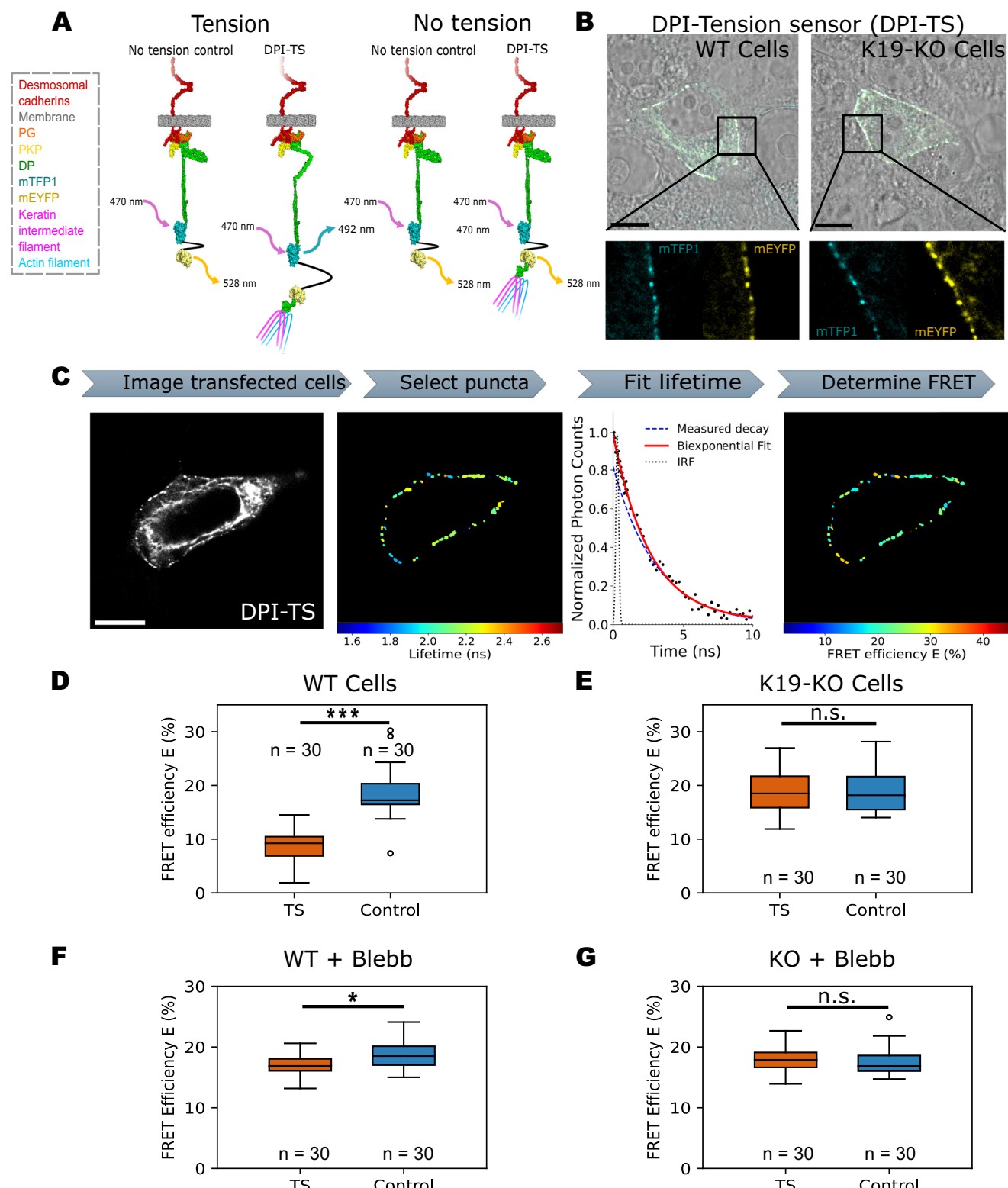

transmitted to desmosomes in a KIF-dependent manner, influencing the mechanical tension experienced by DP.

## DP undergoes force-dependent conformational changes in primary cardiomyocytes

Next, to investigate if DP undergoes similar conformational changes in a non-epithelial system, we studied actomyosin force-dependent DP conformational changes in primary neonatal mouse cardiomyocytes

(CMs). We used neonatal CMs because they (1) organize myofibrils and contract spontaneously and (2) reestablish cell-cell junctions in culture that recapitulate many organizational and architectural hallmarks of the intercalated disc (ICD)−the specialized adhesion structure that connects mature CMs[35–37]. In neonatal cell-cell contacts, adherens junctions, which serve as the primary sites of actomyosin anchorage[36], and desmosomes, which maintain tissue integrity[35,37], are located adjacent to each other. Given this mechanically active junctional

**Fig. 6 | Desmoplakin is mechanically loaded in WT cells but not in K19-KO cells.**
**A** Schematic illustration of the DPI FRET tension sensor (DPI-TS) and no-tension control. The tension-sensing module consists of a linker peptide flanked by mTFP1 (donor fluorophore) and mEYFP (acceptor fluorophore). In the absence of tension, the FRET pair stays together in both constructs. Under tension, the FRET pair is pulled apart in DP-TS, but remains together in the no-tension control, which lacks the keratin-binding C-terminal region. **B** Representative images of WT and K19-KO cells transfected with DP-TS. The overlay of fluorescent signals of mTFP1 (cyan) and mEYFP (yellow) with the bright-field image shows the proper localization of DPI-TS at cell-cell contacts in both WT and K19-KO cells. Scale bar: 10 μm. The zoomed-in images reveal the formation of distinct DP puncta along the cell border. **C** Illustration of the workflow for FRET-FLIM analysis. First, lifetime data images were acquired from transfected cells in fixed samples. Scale bar: 10 μm. Next, individual DP puncta were manually selected as regions of interest along the cell border. The fluorescence lifetime for each DP puncta was determined by fitting a bi-exponential model to distinguish true lifetime measurements from the effects of the instrument response function (IRF). Finally, FRET efficiency (E) was calculated from the lifetime measurements as described in the methods. **D** Quantification of FRET efficiency for DP-TS and no-tension control constructs in WT cells. The median FRET efficiency of the tension sensor was significantly lower than the control, indicating that DP is mechanically loaded in WT cells. $n = 30$ images, $N = 3$, $P = 3.01E{-}14$, ***$P < 0.001$. **E** Quantification of FRET efficiency for DP-TS and no-tension control constructs in K19-KO cells. The median FRET efficiency of DP-TS was indistinguishable from that of the control, suggesting that DP experiences undetectable tension in K19-KO cells. $n = 30$ images, $N = 3$, ns, $P > 0.05$. **F** Quantification of FRET efficiency for DP-TS and no-tension control constructs in WT+Blebb cells. The median FRET efficiency of the tension sensor increased to a level similar to that of the control, indicating that Blebbistatin treatment significantly reduced tension at desmosomes in WT cells. $n = 30$ images, $N = 3$, $P = 1.23E{-}2$, *$P < 0.05$. **G** Quantification of FRET efficiency for DP-TS and no-tension control constructs in KO+Blebb cells. The median FRET efficiency of DP-TS was indistinguishable from that of the control, suggesting that DP experiences undetectable tension in KO+Blebb cells. $n = 30$ images, $N = 3$, ns, $P > 0.05$. All boxplots show median, 25th and 75th percentile with whiskers reaching the last data point within 1.5× interquartile range. Data points outside this range are plotted individually as outliers. The number of data points n represents the number of images for three independent experiments. Statistical significance for all the FRET efficiency measurements is determined using linear mixed-effects model as described in the methods.

architecture, primary CMs offer a physiologically relevant model to test whether the force transmission mechanism we observed in MCF7 cells is conserved in heart tissue. Furthermore, since CMs lack KIFs and instead express the intermediate filament protein desmin[38], CMs allow us to directly test if actomyosin forces induce DP conformational changes independent of K19.

In immature CMs, actomyosin filaments lack a dominant orientation. However, as CMs mature, myofibrils begin to align and desmosomes start to concentrate at CM termini (i.e., the axial membrane where myofibrils terminate), although some desmosomes remain along the lateral membrane, which runs parallel to the myofibrils (Fig. 7A). Consequently, immunostaining for Dsg2, DP, and F-actin revealed desmosomes localized along cell-cell junctions aligned either perpendicular (Fig. 7B-i, axial alignment) or parallel (Fig. 7C-i, lateral alignment) to contractile myofibrils. Notably, the characteristic DP "railroad track" pattern—a hallmark of mature desmosomes—was clearly observed (inset of Fig. 7B-i and 7C-i). Interestingly, axially aligned desmosomes, which are adjacent to myofibril-coupled adherens junctions and presumably experience large contractile actomyosin forces, had a jagged morphology and appeared wider (Fig. 7B-i). In contrast, laterally aligned desmosomes, which are likely exposed to lower tangential actomyosin forces, were arranged more linearly and appeared more compact (Fig. 7C-i).

Though primary CMs can beat in culture, contraction ability and coordination are not uniform across a given culture. Thus, fixation arrests cells in both contractile and noncontractile states, and we imaged differences in desmosome conformation using STED microscopy. To test whether the width of axially and laterally aligned desmosomes reflects differences in actomyosin tension, we treated CMs with Blebbistatin to inhibit actomyosin contractility. Following Blebbistatin treatment, a mixed population of axially aligned desmosomes was observed. In addition to stretched desmosomes, we observed more compact, flattened desmosomes in the Blebbistatin-treated group (Fig. 7B-ii). To quantify these observations, we measured the distance between opposing DP C-termini (DPC-DPC distance) in DMSO-treated control (CM) and Blebbistatin-treated CMs (CM+Blebb). Consistent with our hypothesis, axially aligned desmosomes in untreated CMs displayed significantly greater width than those in CM +Blebb cells (227 ± 62 nm for CM and 178 ± 47 nm for CM+Blebb; Fig. 7D). Moreover, the broader distribution of desmosome widths in the DMSO group likely reflects heterogeneous contraction states at the time of fixation, whereas Blebbistatin-treated CMs were uniformly relaxed, resulting in more consistent and compact junctions. In contrast to the axially aligned desmosomes, the laterally aligned

desmosomes showed no detectable change in desmosome width upon Blebbistatin treatment (Fig. 7C-ii). Quantitative analysis confirmed this observation, with no significant difference in desmosome width between CM and CM+Blebb groups in laterally aligned junctions (172 ± 44 nm for CM and 171 ± 47 nm for CM+Blebb; Fig. 7E). Together, these findings are consistent with the mechanical pathway observed in epithelial cells. Desmosomes aligned along the contractile axis in CMs, like those in MCF7 WT cells, are subjected to high levels of tension and adopt an elongated morphology. In contrast, laterally oriented desmosomes in CMs, analogous to those in K19-KO MCF7 cells, bear lower tension and maintain a more compact organization.

To test if DP in primary CMs undergoes a conformational change under contractile forces, we stained fixed primary CMs with an antibody targeting the N-terminus of DP and measured the distance between opposing DP N-termini (DPN-DPN) in CM and CM+Blebb cells. There was no significant difference in DPN-DPN distance between CM and CM+Blebb conditions in both axially aligned desmosomes (138 ± 36 nm for CM vs. 141 ± 36 nm for CM+Blebb; Fig. 7F) and laterally aligned desmosomes (129 ± 30 nm for CM vs. 127 ± 31 nm for CM +Blebb; Fig. 7G). These results suggest that actomyosin-generated changes in desmosome width resulted from DP conformational changes. Moreover, the narrow and consistent spread of DPN-DPN distances across both treatment groups suggests that the flexibility of DP itself contributes to the broader variability observed in the axially aligned DPC-DPC distances, suggesting that DP adopts a spectrum of conformations in response to different mechanical states. Together, these findings further support our proposed mechanical model: actomyosin contractile forces can be transmitted to desmosomes in primary CMs, inducing structural changes of desmosomes, possibly through DP conformational changes. This mechanism mirrors what we previously observed in MCF7 cells and suggests a conserved role for DP in mediating force-dependent desmosome remodeling in mechanically active tissues.

## Discussion
Based on our data, we propose a model where DP adopts a compact (closed) conformation in the absence of tension and an extended (open) conformation in the presence of high actomyosin forces (Fig. 7H). Several recent studies have highlighted the potential mechanosensitive role of DP in desmosomes. These studies demonstrate that DP experiences tension when cells are subjected to mechanical pulling[32] and show that the DP-KIF linkage is essential for resisting mechanical strain[12,39]. Building on these findings, we demonstrate that DP functions as a mechanosensor by potentially undergoing

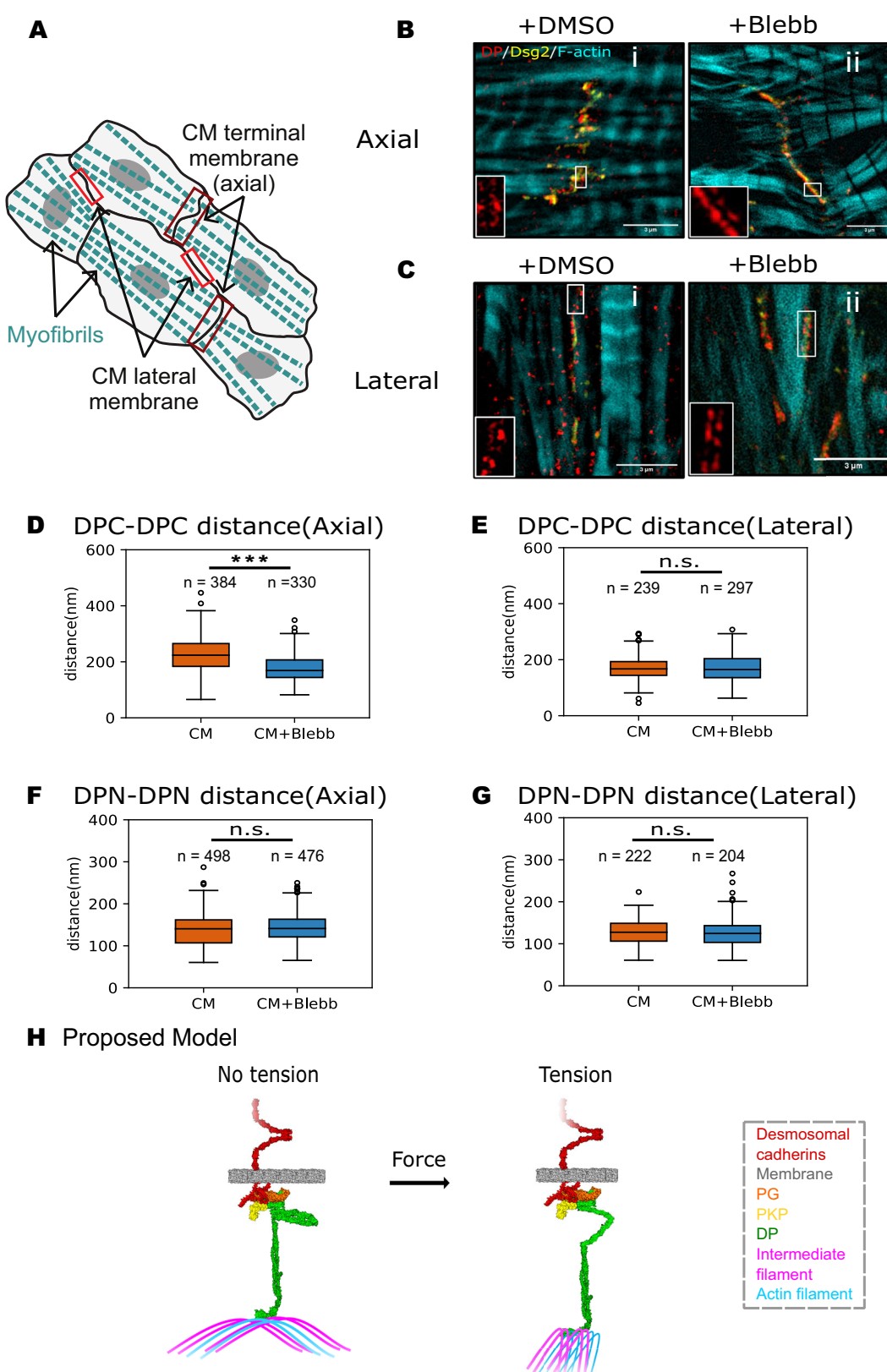

**H** Proposed Model

a conformational change in response to alterations in cytoskeletal tension. We show that these force-induced DP conformational changes occur in both epithelial MCF7 cells and primary CMs, suggesting that DP mechanosensitivity is conserved and occurs across different cell types. The DP conformational change described in our study is analogous to conformational changes in α-catenin, associated with classical cadherins in adherens junctions[40–42]. In the case of α-catenin, tension from the actin cytoskeleton induces a conformational change that opens its central M-domain, exposing cryptic binding sites for vinculin and other actin-binding proteins[40–42]. It is possible that the tension-dependent elongation of DP also exposes cryptic binding sites for junctional proteins that signal and remodel the desmosome and

**Fig. 7 | Desmoplakin undergoes force-dependent conformational changes in primary cardiomyocytes. A** Schematic of polarized CMs. Myofibrils terminate at the axial membrane. The lateral membrane runs parallel to the myofibrils. **B** Representative STED images of axially aligned desmosomes in CMs. The CMs were treated with DMSO (CM, i) or Blebbistatin (CM+Blebb, ii), and immunolabeled for DPC (red), Dsg2 (yellow), and F-actin (cyan). Scale bar: 3 μm. Insets display characteristic DP "railroad track" patterns at the junction. **C** Representative STED images of laterally aligned desmosomes in CMs treated with DMSO (CM, i) or Blebbistatin (CM+Blebb, ii). Scale bar: 3 μm. Insets display DP "railroad track" patterns. **D** Quantification of axially aligned desmosome widths (DPC-DPC distance) in CM and CM+Blebb cells. $n = 384$ (CM), 330 (CM+Blebb). Two-sided Mann–Whitney's U test; $P = 9.813E-28$, ***$P < 0.001$. **E** Quantification of laterally aligned desmosome widths (DPC-DPC distance) in CM and CM+Blebb cells. $n = 239$ (CM), 297 (CM+Blebb). Two-sided Mann–Whitney's U test; ns, $P > 0.05$.

**F** Quantification of axially aligned desmosome widths (DPN-DPN distance) in CM and CM+Blebb cells. $n = 498$ (CM), 476 (CM+Blebb). Two-sided Mann–Whitney's U test; ns, $P > 0.05$. **G** Quantification of laterally aligned desmosome widths (DPN-DPN distance) in CM and CM+Blebb cells. $n = 236$ (CM), 187 (CM+Blebb). Two-sided Mann–Whitney's U test; ns, $P > 0.05$. Desmosomes in (**D**−**G**) were analyzed from CMs isolated from 20 to 24 pups total from two separate preps. **H** Model for force-induced DP conformational change in desmosomes. When desmosomes experience no tension, DP adopts a folded (closed) conformation. Under mechanical stress, the flexible DP plakin domain unfolds to an extended (open) conformation. All boxplots show median, 25th and 75th percentile with whiskers reaching the last data point within 1.5× interquartile range. Data points outside this range are plotted individually as outliers. The number of data points n represents the number of line scans across the desmosomes.

intermediate filaments. The tension-dependent extension of DP could also serve as a damper of mechanical stress, maintaining mechanical homeostasis within the tissue.

Our data suggest that, in addition to actomyosin forces, conformational changes in DP are also mediated by changes in actin organization. Upon K19 loss, we observed a marked reduction in F-actin filament enrichment at desmosomal junctions (Fig. 3). This suggests that K19 plays a critical role in organizing the actin cytoskeleton and facilitating the coupling of actomyosin-generated forces to adhesive sites. The diminished junctional localization of F-actin in K19-KO cells likely results in reduced mechanical tension transmitted to desmosomes, thereby weakening force integration at cell-cell contacts. Loss of K19 also alters the organization of keratin filaments, which in turn affects the efficiency of actomyosin force transmission to desmosomes. We show that in the K19-KO cells, K8/K18 filaments are oriented cortically, almost parallel to the desmosome, resulting in tangential actomyosin forces that cannot stretch DP. In contrast, since KIFs are oriented radially in WT cells, desmosomes are exposed to normal forces, which can induce an axial expansion in DP. Indeed, using FRET tension sensors, we demonstrate that DP in WT cells are exposed to higher tension than DP in K19-KO cells. We confirmed that the tension-dependent widening of desmosomes was not due to differences in the expression levels of desmosomal or cytoskeletal proteins between the two cell lines by comparing the amounts of essential desmosomal and cytoskeletal components in both WT and K19-KO cells (Fig. S1). These results indicate that the observed changes in desmosome organization under force are not due to altered protein expression levels but rather the result of modified mechanical conditions driven by the cooperative interactions between KIF and the actomyosin network.

Our experiments in primary CMs revealed that DP was elongated in actively contracting cells, but more compact in Blebbistatin-treated, relaxed cells. Thus, DP responds to changes in mechanical force in both epithelia and CMs, and we hypothesize that DP functions as a mechanosensitive desmosome component in all load-bearing tissues. Furthermore, since healthy CMs lack keratin filaments, DP conformational changes are either directly driven by actomyosin forces or indirectly regulated through desmin, the primary intermediate filament in CMs. Since K19 in MCF7 cells likely transmits actomyosin-generated forces to desmosomes, larger DP conformational changes were measured when KIFs were oriented roughly perpendicular to the desmosome. Analogous to MCF7 cells, we found that DP conformational changes in CMs also depended on myofibril orientation. Desmosomes aligned along the contractile axis in CMs adopted an elongated morphology, while desmosomes oriented parallel to myofibrils had a more compact organization.

Earlier studies using FRET-based tension sensors in resting MDCK cells revealed a small amount of tension across Dsg2 but not across DP[32,43]. To validate this finding, we also measured DPC-DPC distances in resting MDCK cells. Our measurements revealed that the desmosome width in MDCK cells is comparable to that observed in K19-KO cells, implying that DP in resting MDCK cells indeed do not undergo elongation (Fig. S17). Moreover, modulating actomyosin contractility in MDCK cells, using Calyculin A to enhance tension and Blebbistatin to reduce tension, changed desmosome organization. Specifically, desmosomes became more extended under high-tension states and adopt a more compact morphology under low-tension conditions (Fig. S18), suggesting that tension-dependent DP elongation is not limited to MCF7 cells and CMs. We note that although Calyculin A is widely used to enhance actomyosin contractility in cultured cells[44,45], it broadly inhibits phosphatases and may therefore influence other phosphorylation-dependent pathways, including those regulating the cytoskeleton or desmosomal protein function.

Additionally, FRET-based tension measurements in MCF7 cells, using the same DPI constructs as in previous studies, revealed high levels of tension across DP in WT cells but no detectable tension in K19-KO cells. Upon Blebbistatin treatment, tension levels decreased, as indicated by an increase in FRET efficiency. Notably, this change was observed only in WT cells, since K19-KO cells maintained high FRET efficiency regardless of treatment, suggesting that in the absence of K19, force transmission to desmosomes was already impaired and could not be further reduced by actomyosin inhibition. These FRET-based tension measurements, along with the DPC-DPC distance data, support a correlation between desmosomal tension level and desmosome width, demonstrating that wider desmosomes result from higher tension across the desmosomes.

Previous rotary shadow imaging of DPI reported an average C-to-N-terminal distance of 162 nm[46]. In contrast, transmission EM imaging of desmosomes in bovine nasal epidermis reported a DP length of ~42 nm[47] while super-resolution dSTORM microscopy in human epidermal keratinocytes measured DP lengths of ~54 nm[22], which suggests that DP is tilted at an angle rather than being strictly perpendicular to the plasma membrane. In agreement, our STED measurements reveal a DPI length, based on the antibody-labeling, of ~66 nm in WT MCF7 cells and 33 nm in K19-KO cells. Similarly, in primary CMs, we measure a DPI length of ~45 nm in axially aligned desmosomes and 22 nm in laterally oriented desmosomes. Importantly, since we measure relative distance changes between the N- and C- termini of DP, our measurement of force-induced DP conformational changes remains valid irrespective of the tilted orientation of DP.

Furthermore, based on super-resolution microscopy, a model has recently been proposed where DP increases its tilt as desmosomes mature and become more adhesive, such that narrower and longer desmosomes correspond to stronger adhesive states[22,48]. However, our data indicate that the narrower desmosomes in K19-KO cells are shorter and less adhesive than WT MCF7 cells. We compared the intercellular adhesion of WT and K19-KO cells using a dispase assay, where confluent sheets of cells were detached from the underlying substrate and then subjected to mechanical stress (Fig. S19A). Since cell-cell adhesion is mediated via both Ca²⁺-dependent classical

cadherins[49] and desmosomes, which adhere in Ca$^{2+}$-independent and Ca$^{2+}$-dependent modes[50], we chelated Ca$^{2+}$ prior to application of mechanical stress as previously described[51,52]. Since chelating Ca$^{2+}$ is expected to disrupt both classical cadherin and Ca$^{2+}$-dependent desmosome adhesion, our assay essentially evaluated intercellular adhesion mediated mainly through the Ca$^{2+}$-independent desmosomes. Our data demonstrated that while K19-KO cell sheets fragmented more than WT cells, K19-GFP cell sheets generated a similar number of fragments compared to WT cell sheets (Fig. S19B). These results suggest that in MCF7 cells, wider desmosomes (which are observed in WT and K19-GFP cells) do not correspond to weaker adhesive states. Furthermore, since the proposed model also suggests that desmosomes lengthen as they become more adhesive, we measured the DP length along the cell borders in WT and K19-KO cells (i.e. $\frac{desmosome\ length}{cell}$). In contrast to the proposed model, our results showed a slight decrease in DP length in the weakly adhesive K19-KO cells compared to the strongly adhesive WT cells (Fig. S20). These results support our findings that changes in desmosome width are a direct result of DP conformational changes, which are driven by actomyosin contractility and KIF organization. Our findings are also supported by literature demonstrating that keratin deficiency is associated with impaired intercellular adhesion and increased cellular invasiveness[9,53,54].

Previous SMD simulations carried out on the long arm of the plakin domain (PDB code 3R6N[28]) show that the SRs individually unfold when subjected to pulling force, which exposes the SH3 domain[55,56]. Since our simulations are carried out on the entire plakin domain, which includes the flexible hinge between the long and short arms, we were able to capture the transition of the plakin domain from a closed to an open conformation. Importantly, the conformational change we measure is supported by previous SAXS measurements[20]. This transition of the plakin domain from a closed to an open structure also partially exposes the cryptic SH3 domain, which is known to act as a signal transduction adapter crucial in cell invasiveness, migration, and actin reorganization[57]. However, while our structural model offers a plausible explanation for the tension-dependent elongation of DP, it may be an oversimplification. SR domain unfolding is a well-established phenomenon[58,59] and consequently tension-dependent elongation of DP may occur due to a combination of plakin domain opening and partial unfolding of SR domains. Our simulation methodology also has limitations since AlphaFold only provides static structural predictions and SMD simulations, although useful for probing force-sensitive regions, use high pulling forces and probe shorter timescales than those experienced in physiological contexts. Therefore, our proposed close-to-open structural transition remains a working hypothesis, and future studies using techniques like single-molecule force spectroscopy will be essential to validate this model and clarify the mechanistic contributions of various structural elements to force-induced DP elongation.

Many disease-causing mutations in DP are found within the plakin domain of the N-terminal region, affecting residues both hidden within the protein and exposed to the surrounding environment[60]. Specifically, mutations within the SR7-8 region of the plakin domain have been linked to arrhythmogenic ventricular cardiomyopathy (AVC)[19,60]. Importantly, our simulations predict that the SR7-8 region participates in several key hydrogen bonds with SR5-6, which in turn stabilizes DP in its closed conformation. This raises the intriguing possibility that AVC mutations may interfere with the force-induced structural transition of DP from the closed to the open conformation.

## Methods
### MCF7 cell culture and stabilization
MCF7 WT cells (ATCC), MCF7 K19-KO cells, and MCF7 K19-GFP rescued cells were maintained in low-glucose Dulbecco's Modified Eagle's Medium (DMEM) (Gibco) supplemented with 10% Fetal Bovine Serum (FBS) (Life Technologies) and 1% penicillin-streptomycin (Sigma-

Aldrich) in 5% CO$_2$ at 37 °C. The rescued cells were additionally treated with 100 μg/ml hygromycin for selection of K19-GFP. MDCK cells (ATCC) were cultured in low-glucose DMEM (Gibco) cell culture media with 10% FBS and 1% PSK.

The K19-KO cells were generated using CRISPR/Cas9 as previously described[31]. We developed the K19-GFP rescue cell line following protocols described previously[61]. Briefly, K19-GFP was cloned into K19-KO cells from the pEGFP-K19 plasmid into pLenti CMV Hygro (plasmid #17484. Addgene) using the following primers: forward 5'TCAGTC-GACTGGATCCATGGTGAGCAAGGGCGAGG3' and reverse 5'GAAAGCT GGGTCTAGTCAGAGGACCTTGGAGGC3'. Lentiviral supernatants were generated using the pLenti-K19-GFP plasmid. Lentiviral supernatants collected 24 h post transfection were used to infect subconfluent K19-KO cells in three sequential 4 h incubations in the presence of 4 μg/ml polybrene (Sigma-Aldrich). Transductants were selected using a culture medium containing 100 μg/ml hygromycin beginning 48 h after infection. The K19-GFP rescue cells were selected for further stabilization using fluorescence-activated cell sorting.

### Preparation of MCF7 and MDCK samples for STED and confocal microscopy imaging
Approximately 500,000 cells were seeded for all the MCF7 cell lines, and ~250,000 cells were seeded for MDCK cells on collagen-coated coverslips (22 mm, No.1.5) in a single well of a six-well plate and were kept in 5% CO$_2$ at 37 °C for 24 h. A 'calcium switch' was then performed using the protocol described below to eliminate cell clusters and maintain a similar level of desmosome maturation across all samples. To perform a calcium switch, the cells were first rinsed with phosphate-buffered saline (PBS) (Gibco) solution at room temperature, followed by incubation in 4 mM EGTA (Millipore Sigma) and 10% low Ca$^{2+}$ Fetal Bovine Serum (FBS) in Ca$^{2+}$-free DMEM for 30 min. Following EGTA treatment, the cells were switched to a DMEM medium with 2 mM CaCl$_2$ for 24 h to trigger the formation of mature desmosomes.

Next, cells were fixed either with ice-cold 1:1 methanol and acetone for 10 min (for keratin and DP staining) or with 3% PFA and 0.3% Triton X-100 for 10 min at room temperature (RT) (for F-actin staining). Samples were rinsed with PBS three times and blocked in 1% Bovine Serum Albumin (BSA) overnight at 4 °C. Samples were then incubated with primary antibody (1:200) for 1 h at RT, followed by incubation with secondary antibody (1:600) for 30 min at RT. The samples stained with fluorescently labeled phalloidin (1:400) were incubated with phalloidin for 30 min at RT in the dark. Samples were washed three times with PBS after each antibody incubation. Lastly, samples were mounted to the glass slides using the ProLong Diamond Antifade Mountant (Life Technologies) and stored at 4 °C in the dark.

### Drug treatments of MCF7 and MDCK cells
Cells were cultured as previously described. For the blebbistatin treatment, cells were treated with 50 μM (±) blebbistatin (Millipore Sigma) for 24 h. Blebbistatin was dissolved in DMSO and diluted in the low-glucose DMEM (Gibco) supplemented with 10% FBS (Life Technologies) and 1% penicillin-streptomycin (Sigma-Aldrich) media. The control group was prepared with the same protocol but treated with 0.1% DMSO instead. For the Calyculin A treatment, cells were treated with 100 nM Calyculin A (Millipore Sigma) for 15 min. Calyculin A was dissolved in DMSO and diluted in the low-glucose DMEM (Gibco) supplemented with 10% FBS (Life Technologies) and 1% penicillin-streptomycin (Sigma-Aldrich) media. The control group was prepared with the same protocol but treated with 0.1% DMSO instead.

### Primary cardiomyocyte isolation, culture, and sample preparation
All animal work was approved by the University of Pittsburgh Division of Laboratory Animal Resources. Primary cardiomyocytes were

isolated from Swiss Webster mice (Taconic Biosciences) as described[62]. Mice were housed in the following conditions:12-h light/12-h dark cycle, 18–23 °C ambient room temperature, and 40–60% humidity. Cardiomyocytes were plated onto 35 mm MatTek dishes with 10 mm insets coated with Collagen Type I. Cardiomyocytes were plated in plating media: 65% high glucose DMEM (Thermo Fisher Scientific), 19% M-199 (Thermo Fisher Scientific), 10% horse serum (Thermo Fisher Scientific), 5% FBS (Atlanta Biologicals) and 1% penicillin-streptomycin (Thermo Fisher Scientific). The media was replaced 20–24 h after plating with maintenance media: 78% high-glucose DMEM, 17% M-199, 4% horse serum, 1% penicillin-streptomycin, 1 mM AraC (Sigma), and 1 mM Isoproterenol (Sigma). Cardiomyocytes were cultured in maintenance media for 2 days (72 h post plating), then treated with maintenance media supplemented with 100 µM blebbistatin (dissolved in DMSO) or DMSO alone (negative control) for 1 h at 37 °C. After incubation, cardiomyocytes were fixed in 3% electron microscopy (EM) grade paraformaldehyde (PFA, Electron Microscopy Sciences) in PBS (Ca²⁺, Mg²⁺), 100 mM sucrose, and 0.3% Triton X-100 for 10 min at room temperature. After fixation, cells were washed 3X in PBS (Ca²⁺, Mg²⁺) and then stored in 1% BSA in PBS (Ca²⁺, Mg²⁺) at 4 °C until staining.

### Antibodies

The following primary antibodies were used: human anti-Dsg2 (MAB947, R&D systems), rabbit anti-Dsg2 (21880-1-AP, Proteintech), rabbit anti-DPC antibody (A303-356A, Bethyl Lab), rabbit anti-DPN antibody (25318-1-AP, Proteintech), chicken anti-GFP (600-901-215, Rockland), mouse anti-K19 antibody (A53-B/A2, Santa Cruz Biotechnology), mouse anti-K8 antibody (MA1-06318, ThermoFisher), mouse anti-K18 antibody (MA1-19047, ThermoFisher), control mouse IgG (sc-2025, Santa Cruz Biotechnology), and mouse anti-actin antibody (66009-1-Ig, Proteintech). The following secondary antibodies are used: anti-chicken AF-488 (Invitrogen), anti-mouse AF-594 (Invitrogen), and anti-rabbit AF-647 (Invitrogen). Phalloidin AF-488 (Invitrogen) and AF-568 (Invitrogen) was used for F-actin staining.

The following antibodies were used for the Co-IP and western blot experiments: Anti-Keratin 19 (A53-B/A2), anti-plakophilin-3 (E-10), anti-plakophilin-1 (10B2), anti-desmocolin-1 (A-4), anti-desmocolin-2/3 (7G6), anti-myosin-10 (A-3), and control mouse IgG (sc-2025) from Santa Cruz Biotechnology, anti-Keratin 8 (17514-1-AP), anti-Keratin 18 (10830-1-AP), anti-actin (66009-1-Ig), anti-desmoglein-2 (21880-1-AP), anti-plectin (29170-1-AP), anti-tubulin (66031-1-Ig), and anti-desmoplakin (25318-1-AP) from Proteintech. The following secondary HRP conjugated antibodies were used: anti-mouse-HRP (A4416), and anti-rabbit-HRP (A6154) from Sigma Aldrich.

### STED imaging acquisition and processing

Immunofluorescence images were acquired at RT using a Leica TCS SP8 STED 3X confocal microscope (Leica Microsystems) equipped with a 100-x oil immersion objective (HC PL APO CS2), white light laser for excitation lines between 470 and 670 nm, and depletion laser lines at 592, 660, and 775 nm. The acquisition was performed using LasX software, followed by deconvolution in Huygens Pro software (Scientific Volume Imaging). Post-acquisition processing, including cropping and resizing, color change in the lookup table, adjustment of brightness and contrast, and inversion of grayscale images, was performed using ImageJ software.

### Confocal image acquisition and processing

Immunofluorescence images were acquired using a Leica Stellaris 5 confocal microscope (Leica Microsystems) equipped with a 63x oil immersion objective (HC PL APO CS2). Image acquisition was performed using LasX software, followed by post-processing using ImageJ. During post-processing, images were adjusted with inversion of grayscale, enhanced contrast with 0.3% saturated pixels, and reduced background noise using the despeckle functionality. All images were acquired using the same procedure.

### Distance measurement analysis

We used custom ImageJ macros and Python scripts to quantify the desmosome width along the entire cell-cell border[63]. Since Dsg2 signals colocalize between DP railroad tracks, they were used as the reference to indicate desmosome location along the membrane. Briefly, Dsg2 signals were selected as regions of interest (ROIs), and an algorithm was used to determine the midline of the ROIs along the membrane axis and draw a segment along the midline every 200 nm to obtain the intensity profile of both Dsg2 and DPC or Dsg2 and DPN. Then, Bar, an ImageJ Plugin, was used to determine the location of peaks of different signals in their respective channels along the drawn segment. After obtaining the raw data, we used custom-made MATLAB code[63] to calculate distances for the full desmosome width represented by DPC-DPC or the half desmosome width represented by Dsg2-DPC or Dsg2-DPN. To avoid false signals resulting from background noise or lines not fully drawn over the DP "railroad" track, we filtered out the data that showed less or more than two peaks in the DPC signals along the drawn segment or did not have the Dsg2 signal between the two DPC peaks. Furthermore, in certain desmosomes, the Dsg2 signals were observed as two peaks, as shown in Fig. S21 implying that the desmosome had ruptured at the Dsg2 extracellular region. We eliminated these desmosomes from further analysis and only focused on intact desmosomes with unresolvable Dsg2 signals. For the DP line density measurement, we also used a custom ImageJ macro script for the quantitative analysis[63]. Briefly, we first subtracted the background noise while retaining the DP signals along the cell borders. Then, we skeletonized these signals into segments to obtain the total DP length along cell borders. Lastly, we obtained the DP line density by normalizing the total DP length using the number of cells in the image (i.e., $\frac{\text{desmosome length}}{\text{cell}}$).

### Co-immunoprecipitation

Co-immunoprecipitation was performed as described previously[64]. Cells grown on tissue culture plates were washed three times with 1X cold PBS. Cells were lysed at 4 °C using ice-cold 1% triton lysis buffer (1% Triton X-100, 40 mM HEPES (pH 7.5), 120 mM NaCl, 1 mM EDTA, 1 mM phenylmethylsulfonyl fluoride, 10 mM sodium pyrophosphate, 1 µg/ml each of chymostatin, leupeptin and pepstatin, 10 µg/ml each of aprotinin and benzamidine, 2 µg/ml antipain, 1 mM sodium orthovanadate, 50 mM NaF). Cell lysates were centrifuged to remove cellular debris and protein concentration was determined using Pierce BCA Protein Assay Kit and BSA as standards. Cell lysates were pre-cleared using Protein G Plus-Agarose beads (Santa Cruz Biotechnology) for 40 min at 4 °C while rotating. While rotating, supernatants were incubated with the indicated antibody or IgG control for 2 h at 4 °C. Immuno-complexes were captured using Protein G Plus-Agarose beads and washed 4X using 1% triton lysis buffer supplemented with 300 mM NaCl.

### Gel electrophoresis and immunoblotting

Cells grown on tissue culture plates were washed three times with 1X cold PBS. Cells were lysed at 4 °C using ice-cold 1% triton lysis buffer (1% Triton X-100, 40 mM HEPES (pH 7.5), 120 mM NaCl, 1 mM EDTA, 1 mM phenylmethylsulfonyl fluoride, 10 mM sodium pyrophosphate, 1 µg/ml each of chymostatin, leupeptin and pepstatin, 10 µg/ml each of aprotinin and benzamidine, 2 µg/ml antipain, 1 mM sodium orthovanadate, 50 mM NaF). For immunoblotting, cell lysates were centrifuged at 14,000 rpm (-21,900 × $g$, rotor radius 10 cm) for 5 min at 4 °C to remove cellular debris. Protein concentration was determined using the Pierce BCA Protein Assay Kit following the manufacturer's instructions (WG327385, Thermo Scientific). BSA (Equitech-Bio) prepared in 1% lysis buffer was used as the standard. Aliquots of protein

lysates were resolved using SDS-PAGE and then transferred to nitro-cellulose membranes (0.45 μm) (Amersham Protran; Cytiva) and immunoblotted using the indicated antibodies, followed by horse-radish peroxidase-conjugated goat anti-mouse or goat anti-rabbit IgG (Sigma-Aldrich). Western Lightning Plus-ECL or Amersham ECL Select Western was used as a blotting detection reagent. Signals were detected using iBright™ CL1500 Imaging System (Invitrogen). The immunoblot result for comparing K19 levels among the three cell lines was conducted using a different method, as described previously[65].

### Intensity profile analysis in Z-stack images

Z-stack images were acquired using a Leica Stellaris 5 confocal microscope (Leica Microsystems) equipped with a 63× oil immersion objective (HC PL APO CS2). Image acquisition was carried out using LAS X software, and subsequent image analysis was performed using ImageJ. All images were acquired using identical settings, and the selected Z-range fully encompassed the relevant fluorescence signal regions. During post-analysis, for each image stack, the XY plane with the highest DP signal was identified by generating a maximum intensity projection. A ROI was then defined around a specific cell-cell junction based on this projection. To analyze the junctional distribution of DP and F-actin, we extracted fluorescence intensity profiles along the Z-axis and identified peak intensities along with their corresponding Z positions. To standardize analysis and ensure accurate signal capture, raw peak intensity data and associated Z positions from each junction were processed using a custom Python script[63]. This script filtered data to confirm that the selected Z-range adequately covered the full fluorescence signal for each junction, using a defined intensity threshold to determine the start and end Z positions.

### Keratin filament orientation analysis

Confocal images were acquired as previously described for K8/DP and K18/DP staining. After image acquisition, we utilized a custom ImageJ macro script to determine the global dominant orientation of keratin filaments[63]. Since filament orientation was defined as the angle between the keratin filament and the desmosome axis along the cell-cell junction, we first generated a mask based on DP signals outlining the cell border. This allowed us to define an ROI where the desmosome axis was set to 0°.

For each image, two rectangular ROIs were selected, each extending 2 μm from the cell border into the cytoplasm and having a length matching the desmosome mask. These ROIs were extracted, and the dominant orientation of keratin filaments was determined using the OrientationJ plugin. Specifically, OrientationJ computes the structure tensor, which captures intensity gradients at each pixel and extracts the principal direction of keratin filament orientation, represented as vector fields. The global dominant orientation was then derived by aggregating local orientations across the entire ROI. Consequently, each image provided two data points, each representing the global dominant orientation of desmoplakin-associated keratin filaments at the junction on either side of the cell-cell contact.

### Structure preparation and MD simulation

Plakin domain structure (residues 178-950) was predicted using AlphaFold2[27]. Subsequently, the structure was prepared using PDBFixer[66] and missing hydrogens were added under pH 7.0 conditions. The protein N-terminus was designated with "NH3 + " while "COO-" was assigned to the C-terminus. MD simulations were conducted on the FARM high-performance computing cluster at the University of California, Davis, with GROMACS 2022.3, as previously described[67-69]. The simulations were performed in the OPLS-AA/L force field[70] and TIP4P water model with a 10 Å radius cut-off for Van der Waals and electrostatic interactions. Electrostatic energy calculations employed the particle mesh Ewald method with a 0.16-grid spacing.

At the beginning of the simulation, the plakin domain structure was positioned at the center of a dodecahedral box, ensuring a minimum distance of 1 nm from the boundary for every atom. The box was filled with water molecules and neutralized with charged ions (150 mM NaCl, 4 mM KCl, and 2 mM $CaCl_2$). The system was relaxed with energy minimization and stabilized with equilibration under isothermal-isochoric and isothermal isobaric conditions using a modified Berendsen thermostat and Berendsen barostat. Post-stabilization, a 50 ns MD simulation was performed with 2-fs integration steps, maintaining a temperature of 300 K using a v-rescale thermostat. Equilibration of the protein structure occurred within 20 ns. The number of hydrogen bonds observed between the long arm and the short arm was calculated using gmx hbond.

### Constant-force SMD simulations and analysis

The constant-force SMD simulations were performed as described previously[67-69]. The starting structures for the SMD simulations were the last frame of the corresponding MD simulations. The structures were placed at the side of a rectangular box (23 × 70 × 8 nm box, center at [11.5 nm, 65 nm, 4 nm]). The system, containing ~1,600,000 atoms, was relaxed and equilibrated under isothermal-isobaric conditions using the same protocol as in the MD simulation. The SMD simulations were performed at 310 K temperature using a Nose-Hoover thermostat[71]. During each SMD simulation, we fixed the N-terminus of the plakin domain and pulled the C-terminal region (residues 919–950) along the longest axis of the box with a constant force -830 pN (500 kJ·mol⁻¹·nm⁻¹). The distances between the protein N-terminus and C-terminus were calculated using the gmx pairdist.

### Fluorescence lifetime imaging microscopy (FLIM)

Fluorescence lifetime data were acquired using a Leica TCS SP8 STED 3X confocal microscope, equipped with a pulsed white light laser (NKT Photonics) with wavelengths ranging from 470 to 670 nm, a 63x/1.35 NA oil immersion objective (HC PL APO CS2), and HyD detectors with photon counting capabilities. Images were captured at a scanning speed of 400 Hz with a resolution of 512 × 512 pixels. For each experimental condition, 10 images were acquired from 3 biological replicates.

Cells were cultured overnight on a 35 mm dish with a #1.5 glass-like polymer coverslip (Cellvis) to reach ~70% confluency. They were then transiently transfected with the DPI tension sensor, no-force control, and donor-only control constructs (Addgene), previously generated in an earlier study[32], using ViaFect Transfection Reagent (Promega) in reduced-serum Opti-MEM medium (ThermoFisher). After 8-h incubation at 37 °C with 5% $CO_2$, the culture medium was replaced with low-glucose DMEM (Gibco) supplemented with 10% FBS (Life Technologies) and 1% penicillin-streptomycin (Sigma-Aldrich). Cells were then treated with 1 μg/ml doxycycline to induce gene expression of the DPI constructs through a Tet-On system, which enables conditional gene expression in the presence of dox-ycycline by activating a doxycycline-sensitive transcriptional activator. The following day, cells were fixed with 4% PFA for 10 min at room temperature. After fixation, samples were maintained in Dulbecco's PBS containing $MgCl_2$ and $CaCl_2$ (Sigma-Aldrich) and imaged at RT.

### FLIM-FRET analysis

Fluorescence lifetime data were analyzed using the built-in FLIM Wizard in LAS X software. For each lifetime image, 20–40 ROIs were manually selected, focusing on individual DP puncta. In each ROI, a bi-exponential decay model was fitted to the photon count time data to determine the fluorescence lifetime within each ROI. This model was chosen to mitigate the effects of the instrument response function (IRF) and autofluorescence, which typically result in small lifetime values. FRET efficiency (E) was then calculated using the donor lifetime

in the presence of an acceptor ($\tau_{DA}$) and the mean donor-only lifetime ($\tau_D$), following the equation:

$$E = \left(1 - \frac{\tau_{DA}}{\tau_D}\right) \times 100 \qquad (1)$$

The mean donor-only lifetime was measured using donor-only constructs for each cell line, with WT cells exhibiting a mean lifetime of 2.771 ns and K19-KO cells showing a mean lifetime of 2.784 ns. The mean donor-only fluorescence lifetime was 2.804 ns for WT+Blebb cells and 2.774 ns for KO+Blebb cells. To ensure reliable lifetime fitting, the minimum required photon count was set to 220 photons per image, as determined by the reduction in the spread of the fitted lifetime values.

### Statistical analysis
All statistical tests were performed and all graphs were plotted using custom Python code except those mentioned specifically[63]. In the boxplots, the box represents the 25th and 75th percentiles with the median indicated, and whiskers reach 1.5 times the interquartile range (IQR), defined as the difference between the 25th and 75th percentiles. Data points outside the whiskers are shown as outliers. The number of independent experiments (N) and data points (n) is indicated in each figure legend. Data distribution was first tested based on Shapiro–Wilk's test. Since most data were not normally distributed, non-parametric statistic tests were used for the analysis. Data were analyzed using Mann–Whitney's U test. A multiple-group comparison was performed using the Kruskal–Wallis Test, followed by Dunn's multiple comparison Test. Statistical tests used are specified in the figure legends. Statistical significance was represented at the level of *, $P < 0.05$; **, $P < 0.01$; ***, $P < 0.001$; ns, $P > 0.05$.

For the mean FRET efficiency analysis, statistical tests were conducted in R using a linear mixed-effects model, incorporating a grouped effect to account for puncta derived from the same image as described previously[32]. This approach corrects for statistical dependence among puncta within a single image. Specifically, the model assumes that grouped error across images follows a normal distribution centered around zero. By including this grouped effect, the analysis provides more conservative and robust statistical confidence compared to methods that treat each puncta as an independent data point. In R, the applied linear mixed-effects model was structured as fretEfficiency ~ isTensionSensor + (1|imageNumber), where isTensionSensor was assigned a value of 1 for tension sensor data and 0 for the corresponding truncated control data, while imageNumber uniquely identified each image.

### Reporting summary
Further information on research design is available in the Nature Portfolio Reporting Summary linked to this article.

## Data availability
All processed data have been made available in the manuscript and supporting information. Simulation input files and a coordinate file of the final output have been deposited in a public repository with the identifier https://doi.org/10.5281/zenodo.15151294. Source data are provided with this paper.

## Code availability
Custom code has been deposited at Zenodo with the identifier 16945319.

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

## Acknowledgements

Research was supported by the National Institute of General Medical Sciences of the National Institutes of Health (R01GM121885) to SS; the National Cancer Institute of the National Institutes of Health (R15CA267890-01) to JSC; and the National Heart, Lung, and Blood Institute of the National Institutes of Health (R01 HL127711) to AVK. STED imaging and FRET-FLIM imaging were performed at the UC Davis Advanced Imaging Facility. FACS was performed at the Flow Cytometry Shared Resource, funded by the UC Davis Comprehensive Cancer Center Support Grant (CCSG) awarded by the National Cancer Institute (NCI P30CA093373).

## Author contributions

J.S.C. and S.S. conceived the study. Y.D. and S.S. designed the study. Y.D. performed and analyzed STED experiments, confocal experiments, FLIM-FRET experiments, and dispase assays. A.E. generated the K19-GFP cells and performed co-IP experiments and western blots. B.X. performed and analyzed simulations. Y.H. and A.V.K. harvested, cultured, and fixed primary cardiomyocytes. S.S. supervised the project. Y.D. and S.S. wrote the manuscript with feedback from all authors.

## Competing interests

The authors declare no competing interests.
