## [Peer Review file · Nature Communications]

Actomyosin forces trigger a conformational change in desmoplakin within desmosomes.

Corresponding Author: Professor Sanjeevi Sivasankar

Version 0:

Reviewer comments:

Reviewer #1

(Remarks to the Author)

This study combines super-resolution imaging, FRET-based tension sensors, molecular dynamics (MD) simulations, and biochemical assays to demonstrate that actomyosin forces induce a conformational change in desmoplakin, a key desmosomal protein. The main findings are: (i) Desmoplakin is a force-bearing, mechanosensitive protein that undergoes a force-induced conformational change in response to mechanical stress; (ii) Actomyosin contractility directs mechanical forces along keratin intermediate filaments toward desmoplakin; and (iii) These forces trigger the N-terminal plakin domain of desmoplakin to shift from a closed to an extended (open) conformation. These findings are novel and significant. Most conclusions are well supported by solid experimental evidence; however, some claims would benefit from additional experimental validation.

I strongly recommend publication of this work after either providing new experimental data to strengthen these points or adjusting the tone to present them more cautiously as part of the Discussion section. Detailed comments are provided below.

1. In the second Results section, the authors state: “Quantitative data analysis showed that compared to WT and K19-GFP cells, actin stress fibers in the K19-KO cells were dramatically decreased (Fig. S6B). This suggests that mechanical tension in the K19-KO cells is lower compared to WT and K19-GFP cells. We, therefore, hypothesized that actomyosin-generated forces could be transmitted to the desmosome complex through associations with the KIF network.” However, the rationale behind this interpretation appears weak. Actin stress fiber formation is influenced not only by mechanical tension from actomyosin contractility but also by actin nucleation and regulatory factors. Therefore, the observed decrease in stress fibers in K19-KO cells could reflect more complex changes in cytoskeletal regulation beyond a simple reduction in mechanical tension. The role of the KIF network in this context may involve additional layers of regulation that are not fully accounted for in the current reasoning.

2. In the third Results section, the authors state: “...suggesting that DP was elongated by ~33 nm in the WT cell lines due to higher mechanical tension in these cells.” This observation is well supported by experimental data. However, in the immediately following sentence — “To resolve the mechanistic basis by which DP transitions between open and closed structures...” — the authors attribute the shorter and longer extensions to closed and open structures, respectively. This attribution is not directly supported by experimental evidence.

3. Using AF3 predictions, the authors propose that the closed structure corresponds to a U-shaped conformation of the plakin domains, while the open structure corresponds to an extended state where the U-shape is destabilized under tension. They further support this hypothesis with steered molecular dynamics (MD) simulations. While I appreciate the authors’ efforts, I find that the evidence presented is insufficient to firmly conclude a force-dependent switch from the U-shaped to the extended conformation of the plakin domain. AF3 predictions do not provide information regarding mechanical stability, and steered MD simulations operate at force level and timescales that are not directly comparable to experimental or physiological conditions.

4. Therefore, these strong conclusions should be moved to the Discussion section and framed as speculative. If the authors wish to maintain these claims in the main Results, they must provide direct experimental evidence, such as single-molecule

force spectroscopy, demonstrating the conformational switch from the U-shape to the extended conformation under physiologically relevant forces. Moreover, they should address the possibility of spectrin repeat (SR) unfolding, as it is well established that SR domains can unfold at forces of a few pN (PMID: 32900995; PMID: 30457830) — within the range reported by their F40 tension sensor measurements.

Reviewer #2

(Remarks to the Author)

In this manuscript by Dong et al, the authors present an exciting idea that desmoplakin (DSP), a critical protein in the formation of desmosomes, undergoes conformational changes in response to mechanical loading.

I found this work very interesting and felt it may help to establish DSP as a new mechanically responsive protein, which may impact how the field thinks of desmosomes as mechanically responsive structures similar to concepts that have been established for focal adhesions (through conformationally responsive proteins vinculin and talin).

Although I have no concerns with the quality of the data presented, or the conclusions presented (that there are significant conformational changes in desmoplakin), I however have some concerns that this work stops short of identifying how these conformational changes occur and why these conformations are biologically significant. As such, I wish to suggest that the authors better explore some aspects of this to increase the impact of this exciting work. I have listed 3 potential ideas to explore, but I think if the authors were to accomplish only 1 (or even something different that is suggested by other reviewers), this would resolve my concern that the authors did not adequately demonstrate that the DSP conformational changes are biologically significant.

Potential ideas for improving impact:

1. The authors perform computational simulations to show how desmoplakin may be unfolding. This is a good start, but it would have been more exciting if the authors could experimentally demonstrate that these specific regions of the protein are indeed unfolding. The obvious idea would be to develop point mutations where the protein is stabilized either in the folded or unfolded conformation. There may be alternate ideas such as removing this entire region to create a shorter DSP that doesn't change in length. This reviewer is acutely aware of the challenges of working with DSP, as its larger size that makes cloning efforts more challenging, including transfection/transduction efforts to re-express mutant DSP in cells. However, if the authors were able to identify some methods to alter these conformational changes this would be a huge impact to the field and provide important information and tools to other researchers (thereby increasing the impact of this work).
2. An alternate, unexplored question, is do the conformational changes influence signaling or protein binding at the desmosome adhesion? I think this group could easily explore this using BioID or other proximity-based interactome labeling to see if the folded vs unfolded conformation has altered protein binding. If successful, this would increase the impact of their work by showing that the conformational changes are functionally important by altering protein binding at desmosomes. This could also help address #1 if the authors could design the proximity labeling approach to determine if the changes in protein binding were in proximity to the conformational site.
3. The last unexplored area (in this reviewer's opinion) is why keratin 19 influences cellular contractility. This is likely beyond the focus of this paper. However, I think if the reviewers were not able (or chose not) to accomplish #1 and #2, providing insight into how keratin networks regulate mechanical forces at cell-cell adhesions would increase the impact of their study.

Additional concerns and suggestions:

1. I found the statistical labeling of the graphs (1i, 2e, 2g) to be incomplete and unclear. I think the authors want to compare group 1 vs 2, 2 vs 3, and then 1 vs 3. Thus there should be 3 comparisons made (only 2 are). 2e is especially confusing as only 1 set of *** are shown but two sets of lines for comparisons are drawn. I consider this only minor, as to this reviewer the statistical differences are quite clear and support the overall conclusions.
2. I would caution the reviewers on possibly overinterpreting the co-IP studies, as the interactions of K19 with these proteins may be occurring outside of the desmosomes. It is known that desmosomes are very insoluble and thus the interactions observed in these assays are elsewhere in the cell. I think the co-IPs are good to include, but the authors may consider making a comment that the interactions detected could be occurring elsewhere in the cell.
3. I liked the use of MDCK as a lower force (K19 free) system to show that DSP is not in the more open conformation. Did the authors try any efforts to increase contractility in these cells (stretch or using the Rho Activating peptide available from Cytoskeleton) to show if increased force at desmosomes changes the conformation of DSP? I recognize the authors did something similar with Calyculin A in the K19 KO cells. I would suggest they still try an experiment to increase forces in the MDCK cells to determine if force-driven DSP conformational changes occur also in another cell line (MDCK) or determine if it is limited only to MCF7. I also wonder if there could be non-specific effects of calyculin A in these responses, given its broad inhibition of phosphatases. Although it does lead to activation of myosin, it is also conceivable that it increases phosphorylation of desmosomal or keratin-associated proteins to change the conformation of DSP.
4. Can the authors include representative images in Figure 4 for the calyculin A treatments, similar to those shown in 4C, D, H, and I?

Reviewer #3

(Remarks to the Author)

The manuscript by Dong et al. uses super-resolution imaging, computer simulations and a novel FRET-based tension sensor for desmoplakin to show that actomyosin-mediated forces contribute to conformational changes in desmoplakin. The authors suggest that forces induce a mechanosensitive conformational change in the N-terminal plakin domain of

desmoplakin, changing it from a folded to an extended conformation.

The manuscript is very well written and the experiments are well presented and analysed, but essential experiments to provide evidence for the claims made are missing. In addition, while it is interesting that desmoplakin appears to be under force, there is very little additional mechanistic and molecular insight into the functional consequences of the hypothesised conformational changes.

Main weaknesses:

1) The authors suggest that a mechanosensitive conformational change in the N-terminal plakin domain of desmoplakin is responsible for the widening of tramlines in desmosomes that occurs under tension. While this may be plausible and the section looks interesting as the authors have identified some key residues that may be involved in opening this hinge, this model of extended conformation needs to be experimentally validated. It would be essential to show that removing the hinge region or introducing mutations that inhibit the conformational switch abolishes the widening of the tramline.

2) Why did the authors use calyculin A to induce actin stress fibres? This can have so many different side effects. Activating RhoA is the direct way to induce actomyosin activation. However, the effect on force generation needs to be shown in a direct way. In addition, the actin close to the desmosomes, even at high resolution as shown in Figure 2b, is not very convincing because it is not clear whether they would interact and whether forces are exerted at these sites. In fact, it is very unlikely that they do so at these sites, since the cortical actin ring would rather be determined by adherens junctions in the vicinity. Also, in this case, the keratins are not oriented perpendicular to the desmosomes and therefore represent desmosomes under force.

3) The western blots are not convincing. (i) they consist of excised regions and do not show the whole blot with appropriate MW indications and (ii) both keratin and actin are so abundant in cells with multiple indirect cross-talk possibilities. To show a direct desmosomal link, the authors should perform desmoplakin IPs in cells with and without K19 and with desmoplakin mutants that cannot bind to intermediate filaments (e.g. with those that were used in tension sensor experiments).

4) Just showing a tension sensor and possible conformational changes in a protein is rather sparse information. The authors state in their discussion that "it is likely that force-induced opening of DP also exposes cryptic binding sites for junctional proteins that signal and remodel the desmosome and intermediate filaments". Without showing this, or exactly how actin contributes to the hinge switch, or whether perturbing the switch has functional consequences, there is little conceptual advance in the field.

5) The most obvious control experiment for force contribution and tension sensors, i.e. blebbistatin treatment to inhibit actomyosin, has not been done. If these tension sensors work, then there is the opportunity of using them in living cells \pm induction & inhibition of forces. The strength of intramolecular FRET probes and authors should be able to show the transition from high to low force and vice versa including changes in FRET changes in living cells.

6) If cells still have K8/18, why does DP remain in a closed conformation in K19 KO cells (line 300)? The authors speculate that the loss of K19 alters the organisation of keratin filaments to become more cortical and transmit less force through DP, but their keratin orientation data is not strong enough to support this at present. It would be interesting to see the effect of K18 KO for comparison and to test the hypothesis that K19 orients filaments radially in WT cells.

7) There is very little mechanistic insight into the contribution of the different keratins to forces. All keratins appear to interact with actin, but only a subset are efficient force transducers. As there is little mechanistic insight, this section remains purely descriptive and raises many questions rather than providing functional insight. The apparent differences seen in MDCK and MCF7 cells are also difficult to understand and raise additional questions.

Minor points:

- The authors' description of the desmoplakin structure (lines 67-68) seems to me to be a bit off. I believe that the globular head region generally encompasses the entire N-terminus (everything before the rod domain) and includes spectrin repeats 1 and 2 as well as the downstream plakin domain. However, the authors seem to refer to the region before the plakin domain as the globular head domain.

- As a general observation, the authors seem to use subconfluent monolayers of MCF-7 cells throughout the study. I wonder if this is suboptimal for establishing uniform tension across the epithelial sheet. On the other hand, polarised migrating cells, which are likely to exert more force on desmosomes than cells in confluent areas, could be more rigorously analysed for their cytoskeletal appearance (F-actin, keratin and desmosomes), which should change under blebbistatin treatment.

- Figure 4A is oddly cropped in the 'no tension' schematic. The comparison between K8 filaments in WT and K19-KO cells (line 220) is not convincing because the representative images chosen are not comparable. The junctions shown in wild-type cells are shorter, as if formed between two lamellipodia, whereas the K19-KO image shows a long continuous junction between two adjacent cells. This also applies to the K18 images in Figures 4H and should be corrected when published. The authors also fail to show representative images of drug-treated samples. These should at least be included in the Supplements.

- The authors use dispase (line 341) to lift sheets of WT or K19-KO MCF-7 cells and incubate them in EGTA for 1 hour to select for hyperadhesive desmosomes. To my knowledge desmosomes reach only hyperadhesion during prolonged cell culture? Under the conditions in which the cells are cultured in this manuscript, it is likely that they are still Ca-dependent.

The question is whether EGTA has a limited effect on cell-cell adhesion under these conditions and the disruption is due to increased fragility of the intermediate filaments?

Reviewer #4

(Remarks to the Author)

Version 1:

Reviewer comments:

Reviewer #1

(Remarks to the Author)

The authors have thoroughly addressed all of my comments, concerns, and suggestions, and have revised the manuscript accordingly. I support the publication of this work.

Reviewer #2

(Remarks to the Author)

In this revised manuscript the authors have made substantial improvements to the manuscript. In particular, there is new evidence showing that DSP changes in length in cardiomyocytes, as well as when contractility is increased in MDCK cells. This reviewer remains very excited by the new finding that DSP may experience mechanosensitivity through a possible hinge/unfolding region, and that this finding appears to occur in multiple cell types (the addition of the cardiomyocytes further increases the significance/importance of their findings). The authors data establishes that the length is changing dependent on actomyosin contractility.

While I am very curious how this 'hinge' region works and what are the biological consequences of it, I do feel that it would be a disservice to hold back this publication for some time while the authors figured this out. Thus, I believe the scientific community is better served by having these findings published vs waiting for additional information and context.

Reviewer #3

(Remarks to the Author)

Many of the data have improved and are of high quality. However, it is rather unfortunate that the authors have not addressed, in my opinion, the most important question, i.e. the modification of the hinge region to show that the hypothesised conformational change is true. While DP is a large protein, the modern cloning strategies and transfection possibilities should enable these deletions/modifications and expression in cells. While it may be challenging, I don't think it is "immensely challenging" but rather doable. It seems that the authors have done cloning and transfections sufficiently to analyse the existing tension sensors and it is difficult to understand why they cannot do modifications and transfections to verify their hypothesis.

Similarly, the AlphaFold predictions are quite powerful and can be used to design mutations to bolster or disrupt predicted residue interactions in the closed conformation. Again this would be easy to do, particularly on the basis that essential parts of the manuscript are based on modelling and MD simulations. Additionally, it would be nice to see the AlphaFold confidences projected onto their model which would aid in identifying confidently predicted interactions.

I feel that this is not beyond the scope of this manuscript; it is the focal point of the manuscript which should be verified and it provides the real novelty for publication in Nat Comms. Novelty without these data are marginal since DP tension sensors have been published previously showing that it is under mechanical load under specific conditions.

Minor points:

The inclusion of FRET control experiments in the presence of blebbistatin adds weight to the actomyosin mediated force hypothesis. In line with their other experiments, it would have also been useful to see whether the FRET readout implies increased DP tension upon calyculin A treatment.

Response 24 still doesn't justify, in my opinion, why the authors chose to use EGTA at all. Especially for an extended period, one would expect to lose all desmosome mediated adhesion? As far as I am aware, it is not a normal step in the disperse assay.

Reviewer #4

(Remarks to the Author)

I co-reviewed this manuscript with one of the reviewers who provided the listed reports. This is part of the Nature Communications initiative to facilitate training in peer review and to provide appropriate recognition for Early Career

Researchers who co-review manuscripts.

Response to Reviewers

We appreciate all the reviewers' comments and suggestions. We believe that incorporating the reviewer's feedback has significantly strengthened our manuscript.

First, we would like to highlight the major changes we have incorporated.

1. We now include experiments (new figure, Fig. 3) to provide additional mechanistic insights into how keratin-19 (K19) regulates mechanical force on desmosomes. Our new data demonstrates that K19 plays a key role in organizing F-actin filaments at desmosomes and regulating the level of actomyosin forces integrated into the desmosomal complex.
2. To cement our findings that actomyosin forces induce a conformational change in desmoplakin (DP), we now perform super-resolution imaging experiments in primary cardiomyocytes (new figure, Fig. 7). These data demonstrate that DP in cardiomyocytes undergoes an actomyosin force-dependent conformational change. Similar to MCF7 cells, where DP conformational change depended on actin filament orientation, we observe that DP conformational change in cardiomyocytes depends on myofibril orientation. Importantly, these new experiments suggest that the force-dependent DP conformational changes are not limited only to epithelial systems.
3. We have also performed the additional control experiments (Fig. 6F-G, Supplementary Figs. S5, 12, 13, and 18) and analysis requested by the reviewers.
4. We have also edited the manuscript to be more cautious in our interpretations and avoid overextending our findings.

Below is our point-by-point response to the reviewers. The original reviewer comments are highlighted in *blue*, and our response is in black.

Reviewer #1 (Remarks to the Author)

This study combines super-resolution imaging, FRET-based tension sensors, molecular dynamics (MD) simulations, and biochemical assays to demonstrate that actomyosin forces induce a conformational change in desmoplakin, a key desmosomal protein. The main findings are: (i) Desmoplakin is a force-bearing, mechanosensitive protein that undergoes a force-induced conformational change in response to mechanical stress; (ii) Actomyosin contractility directs mechanical forces along keratin intermediate filaments toward desmoplakin; and (iii) These forces trigger the N-terminal plakin domain of desmoplakin to shift from a closed to an extended (open) conformation. These findings are novel and significant. Most conclusions are well supported by solid experimental evidence; however, some claims would benefit from additional experimental validation.

I strongly recommend publication of this work after either providing new experimental data to strengthen these points or adjusting the tone to present them more cautiously as part of the Discussion section. Detailed comments are provided below.

We thank Reviewer #1 for their positive feedback. We have addressed the reviewer's comments below.

1. In the second Results section, the authors state: “Quantitative data analysis showed that compared to WT and K19-GFP cells, actin stress fibers in the K19-KO cells were dramatically decreased (Fig. S6B). This suggests that mechanical tension in the K19-KO cells is lower compared to WT and K19-GFP cells. We, therefore, hypothesized that actomyosin-generated forces could be transmitted to the desmosome complex through associations with the KIF network.” However, the rationale behind this interpretation appears weak. Actin stress fiber formation is influenced not only by mechanical tension from actomyosin contractility but also by actin nucleation and regulatory factors. Therefore, the observed decrease in stress fibers in K19-KO cells could reflect more complex changes in cytoskeletal regulation beyond a simple reduction in mechanical tension. The role of the KIF network in this context may involve additional layers of regulation that are not fully accounted for in the current reasoning.

Response 1: We thank the reviewer for their insightful comment. We agree that stress fiber formation is influenced not only by mechanical tension but also by upstream regulatory pathways, including Rho GTPase signaling and actin nucleation factors. On **lines 159-162**, we now clarify that the reduction in stress fibers in K19-KO cells may also be influenced by broader changes in cytoskeletal regulation beyond a direct loss of mechanical tension. We therefore use actin stress fibers only as an initial indication that K19 may play a role in transmitting actomyosin-generated forces to desmosomes.

2. In the third Results section, the authors state: “...suggesting that DP was elongated by ~33 nm in the WT cell lines due to higher mechanical tension in these cells.” This observation is well supported by experimental data. However, in the immediately following sentence — “To resolve the mechanistic basis by which DP transitions between open and closed structures...” — the authors attribute the shorter and longer extensions to closed and open structures, respectively. This attribution is not directly supported by experimental evidence.

Response 2: The reviewer is correct in pointing out that the attribution of the shorter and longer extensions to “closed” and “open” structural states, respectively, was interpretative and not directly validated by structural experiments.

To address this concern, we have extensively revised the manuscript to be more cautious in our statements. For instance, in the abstract (**lines 30-32**) we clarify that the proposed conformational change is *plausible*. Similarly, in the introduction (**lines 94-96**), we clarify that MD simulations *suggest* a conformational change in the N-terminal plakin domain. We have made similar changes throughout the manuscript to avoid overextending our findings. We further clarify that our working model of DP conformational change is informed by structural predictions and simulations, rather than direct experimental evidence. Specifically, we have reworded the relevant text in the Results section, **lines 231, and 254-256**, to describe the observed DP elongation in neutral terms, without equating it to a definitive open or closed state.

3. Using AF3 predictions, the authors propose that the closed structure corresponds to a U-shaped conformation of the plakin domains, while the open structure corresponds to an extended state where the U-shape is destabilized under tension. They further support this hypothesis with steered molecular dynamics (MD) simulations. While I appreciate the authors' efforts, I find that the evidence presented is insufficient to firmly conclude a force-dependent

switch from the U-shaped to the extended conformation of the plakin domain. AF3 predictions do not provide information regarding mechanical stability, and steered MD simulations operate at force level and timescales that are not directly comparable to experimental or physiological conditions.

Response 3: We thank the reviewer for this thoughtful critique. Given the limitations of AlphaFold predictions and SMD simulations, we believe that our structural interpretation of the “closed” and “open” DP conformation should be presented with greater caution. In the revised manuscript, we have modified the language to frame these interpretations as hypotheses rather than conclusions, as noted in Response 2.

Additionally, we have added text in the Discussion (**lines 527-537**) that explicitly addresses the limitations of AlphaFold predictions and SMD simulations in predicting force-dependent conformational transitions. For instance, we emphasize that our proposed structural model may be an oversimplification since the observed elongation of DP may result from a combination of plakin domain opening via the flexible hinge region and partial unfolding of SR domains. We also clarify that while our structural model offers a plausible explanation for the tension-dependent elongation of DP, there are limitations of this approach since AlphaFold provides static structural predictions. Similarly, SMD simulations, although useful for probing force-sensitive regions, utilize high pulling forces and probes shorter timescales than those experienced in physiological contexts. Therefore, our proposed close-to-open structural transition remains a working hypothesis, and future studies using techniques, like single-molecule force spectroscopy, will be essential to validate this model and clarify the mechanistic contributions of various structural elements to force-induced DP elongation.

4. Therefore, these strong conclusions should be moved to the Discussion section and framed as speculative. If the authors wish to maintain these claims in the main Results, they must provide direct experimental evidence, such as single-molecule force spectroscopy, demonstrating the conformational switch from the U-shape to the extended conformation under physiologically relevant forces. Moreover, they should address the possibility of spectrin repeat (SR) unfolding, as it is well established that SR domains can unfold at forces of a few pN (PMID: 32900995; PMID: 30457830) — within the range reported by their F40 tension sensor measurements.

Response 4: We agree that the mechanistic interpretation of a force-induced conformational change in DP was too speculative for the Results section. As outlined in Responses 2 and 3, we have revised the manuscript to retain only the direct simulation outputs in the Results and have relocated the mechanistic interpretation regarding DP unfolding to the Discussion, where it is now clearly presented as a speculative model. Additionally, as described in Response 3, we have expanded the Discussion to address the limitations of our approach.

Reviewer #2 (Remarks to the Author):

In this manuscript by Dong et al, the authors present an exciting idea that desmoplakin (DSP), a critical protein in the formation of desmosomes, undergoes conformational changes in response to mechanical loading.

I found this work very interesting and felt it may help to establish DSP as a new mechanically responsive protein, which may impact how the field thinks of desmosomes as mechanically responsive structures similar to concepts that have been established for focal adhesions (through conformationally responsive proteins vinculin and talin).

We thank Reviewer #2 for their positive feedback. We have addressed the reviewer's comments below.

Although I have no concerns with the quality of the data presented, or the conclusions presented (that there are significant conformational changes in desmoplakin), I however have some concerns that this work stops short of identifying how these conformational changes occur and why these confirmations are biologically significant. As such, I wish to suggest that the authors better explore some aspects of this to increase the impact of this exciting work. I have listed 3 potential ideas to explore, but I think if the authors were to accomplish only 1 (or even something different that is suggested by other reviewers), this would resolve my concern that the authors did not adequately demonstrate that the DSP conformational changes are biologically significant.

Response 5: We thank the reviewer for their thoughtful and constructive suggestions aimed at increasing the impact of this work. We fully agree that a deeper understanding of the mechanistic basis and biological significance of DP conformational changes would be highly valuable to the field. The reviewer listed three potential ideas but indicated that accomplishing any one of those ideas would resolve their concerns. Following the reviewer's advice, we have now accomplished the third idea that the reviewer suggested (listed in **Response 8**).

Finally, even though the reviewer did not ask for this, we decided to do an additional set of experiments to demonstrate that DP conformational changes are biologically relevant and extend beyond MCF7 cells. As described in **Response 11**, we performed STED imaging experiments in primary cardiomyocytes and demonstrated that DP undergoes an actomyosin force-dependent conformational change. Like MCF7 cells, where DP conformational change depended on actin filament orientation, we show that DP conformational change in cardiomyocytes depends on myofibril orientation.

Potential ideas for improving impact:

1. The authors perform computational simulations to show how desmoplakin may be unfolding. This is a good start, but it would have been more exciting if the authors could experimentally demonstrate that these specific regions of the protein are indeed unfolding. The obvious idea would be to develop point mutations where the protein is stabilized either in the folded or unfolded confirmation. There may be alternate ideas such as removing this entire region to create a shorter DSP that doesn't change in length. This reviewer is acutely aware of the challenges of working with DSP, as its larger size that makes cloning efforts more challenging, including transfection/transduction efforts to re-express mutant DSP in cells. However, if the authors were able to identify some methods to alter these conformational changes this would be a huge impact to the field and provide important information and tools to other researchers (thereby increasing the impact of this work).

Response 6: As suggested by the reviewer, experimentally restricting DP conformational changes via targeted mutagenesis or truncation constructs would indeed provide important mechanistic insight. However, as the reviewer recognizes, these are very challenging experiments given the substantial size of the DP protein and given that crystal structures of only a minuscule fragment of the protein have been solved. Furthermore, as pointed out by Reviewer#1, the structural insights we obtain from AlphaFold predictions and SMD are inherently limited. We therefore believe that the type of structure-function analysis proposed by the reviewer is beyond the scope of our current study. However, we view the reviewer's suggestion as a critical direction for future work and have emphasized the importance of specific domains or residues in DP conformational regulation in **lines 538-545** of the Discussion section.

2. An alternate, unexplored question, is do the conformational changes influence signaling or protein binding at the desmosome adhesion? I think this group could easily explore this using BioID or other proximity-based interactome labeling to see if the folded vs unfolded conformation has altered protein binding. If successful, this would increase the impact of their work by showing that the conformational changes are functionally important by altering protein binding at desmosomes. This could also help address #1 if the authors could design the proximity labeling approach to determine if the changes in protein binding were in proximity to the conformational site.

Response 7: Similarly, we appreciate the reviewer's suggestion to investigate whether DP conformational states influence protein interactions at desmosomes, for example, through proximity-labeling approaches such as TurboID. We agree that mapping the interaction landscape of folded versus unfolded DP could reveal novel regulatory or signaling functions associated with its mechanosensitive behavior. This is particularly intriguing given that the transition of the plakin domain from a closed to an open conformation may expose the cryptic SH3 domain, which is known to function as a signal transduction adapter involved in processes such as cell migration, invasiveness, and actin cytoskeletal remodeling [1]. However, due to the large size of DP, generating cells that stably express DP-TurboID fusion constructs is challenging. While such interactome profiling is beyond the scope of the current study, we consider this a compelling direction for future work to elucidate the functional significance of force-induced DP conformational changes.

3. The last unexplored area (in this reviewer's opinion) is why keratin 19 influences cellular contractility. This is likely beyond the focus of this paper. However, I think if the reviewers were not able (or chose not) to accomplish #1 and #2, providing insight into how keratin networks regulate mechanical forces at cell-cell adhesions would increase the impact of their study.

Response 8: We thank the reviewer for raising this valuable suggestion. To address how keratin networks regulate mechanical forces at cell-cell adhesions, we have now conducted new experiments and included a new **Figure 3** (see below Fig. R1). These findings strengthen our study by directly linking K19 to actin-desmosome coupling at cell-cell junctions.

We acquired Z-stack confocal images of MCF7 WT and K19-KO cells stained for DP and F-actin. Maximum intensity projections were used to define junctional ROIs, from which we generated XZ cross-sections and extracted fluorescence intensity profiles. Notably, F-actin

Figure R1: KIF network regulates F-actin coupling to desmosomes. (A) Representative confocal overlay image of XY maximum intensity projection of DP and F-actin in MCF7 cells. Scale bar: 10 μm . (B) Magnified view of the boxed area in (A) illustrates the selected ROI at a specific cell-cell junction. (C) XZ plane of cross-sectional view of the ROI demonstrates the distribution of DP and F-actin signals along the X and Z axes. (D) Fluorescence intensity profile of DP and F-actin reveals peaks intensities and corresponding Z positions at the junction. (E) Scatter plot of actin peak intensities along the Z axis in WT and K19-KO cells shows generally higher actin peak intensities in WT compared to K19-KO cells. (F) Quantification of actin peak intensities along the Z axis from WT and K19-KO cells. $n = 197$ (WT), 331 (K19-KO); $N=3$. Mann-Whitney's U test; ***, $P < 0.001$. Actin peak intensities along the junction are significantly higher in the WT compared to the K19-KO cells. (G) Scatter plot of DP peak intensities along the Z axis in WT and K19-KO cells shows overlapping distributions between WT and K19-KO cells. (H) Quantification of DP peak intensities along the Z axis from WT and K19-KO cells. $n = 197$ (WT), 331 (K19-KO); $N=3$. Mann-Whitney's U test; *, $P < 0.05$. The DP peak intensities along the junction are slightly greater in the K19-KO compared to the WT cells. (I) Scatter plot of actin/DP peak intensity ratios along the Z axis from WT and K19-KO cells shows clear separation between WT and K19-KO cells. (J) Quantification of actin/DP peak intensity ratios along the Z axis from WT and K19-KO cells. $n = 197$ (WT), 331 (K19-KO); $N=3$. Mann-Whitney's U test; ***, $P < 0.001$. The actin/DP peak intensity ratio along the junction are significantly greater in the WT compared to the K19-KO cells

peaks closely aligned with DP peaks along the Z-axis, suggesting coordinated spatial organization (Fig. R1A–D).

Quantitative analysis revealed significantly higher F-actin peak intensity in WT cells than in K19-KO cells (90.3 ± 41.5 vs. 38.9 ± 27.0 a.u.; Fig. R1E–F), while DP intensity remained comparable (Fig. 3G–H). Importantly, the F-actin/DP intensity ratio was ~3-fold higher in WT than in K19-KO cells (1.16 ± 0.56 vs. 0.38 ± 0.20 ; Fig. R1I–J), indicating that K19 is required for effective F-actin recruitment to desmosomes.

Together, these results support a key role for K19 in organizing F-actin filaments and anchoring them to desmosomes, thereby facilitating actomyosin force transmission at cell-cell junctions. We have presented these results in **lines 184-212** and shown them in **Figure 3**.

Additional concerns and suggestions:

*1. I found the statistical labeling of the graphs (1i, 2e, 2g) to be incomplete and unclear. I think the authors want to compare group 1 vs 2, 2 vs 3, and then 1 vs 3. Thus there should be 3 comparisons made (only 2 are). 2e is especially confusing as only 1 set of *** are shown but two sets of lines for comparisons are drawn. I consider this only minor, as to this reviewer the statistical differences are quite clear and support the overall conclusions.*

Response 9: We apologize for this confusion. We have now revised the figures to clearly display the statistical annotations for all comparisons. The corrected versions for Fig. 1I, 2E, and 2G are provided in Fig. R2 below for your reference.

Figure R2: Revised figures with appropriately labeled statistical annotations. Revised Fig. 1I (left panel), revised Fig. 2E (middle panel), and revised Fig. 2G (right panel).

2. I would caution the reviewers on possibly overinterpreting the co-IP studies, as the interactions of K19 with these proteins may be occurring outside of the desmosomes. It is known that desmosomes are very insoluble and thus the interactions observed in these assays are elsewhere in the cell. I think the co-IPs are good to include, but the authors may consider making a comment that the interactions detected could be occurring elsewhere in the cell.

Response 10: We appreciate the reviewer's insightful comment. To address this point, we have clarified in **lines 152-154**, that the interactions between K19 and desmosomal proteins detected by co-IP may occur outside of desmosomes.

3. I liked the use of MDCK as a lower force (K19 free) system to show that DSP is not in the more open conformation. Did the authors try any efforts to increase contractility in these cells (stretch or using the Rho Activating peptide available from Cytoskeleton) to show if increased force at desmosomes changes the conformation of DSP? I recognize the authors did something similar with Calyculin A in the K19 KO cells. I would suggest they still try an experiment to increase forces in the MDCK cells to determine if force-driven DSP conformational changes occur also in another cell line (MDCK) or determine if it is limited only to MCF7.

Response 11: We thank the reviewer for this thoughtful comment. As suggested by the reviewer, we have now altered the contractility of MDCK cells and demonstrated that changing the force at desmosomes changes the conformation of DP. We used Calyculin A to enhance actomyosin contractility and Blebbistatin to reduce tension. As shown in Fig. R3, desmosomes became significantly more extended under high-tension conditions and more compact under reduced tension, suggesting that tension-dependent DP elongation is not limited to MCF7 cells. These new results are presented in **lines 467-472** of the Discussion Section and shown in **Supplementary Fig. S18**.

Figure R3: Desmosome width in MDCK cells is altered in response to changes in actomyosin forces. Quantification of desmosome width in MDCK cells with Calyculin A treatment (A). n = 357 (MDCK), 500 (+Caly), 434 (DMSO); N = 3. Kruskal-Wallis Test, followed by Dunn's multiple comparison Test; ***, P<0.001; ns, P>0.05. Quantification of desmosome width in MDCK cells with Blebbistatin treatment. Desmosome width in MDCK cells becomes significantly higher upon increased actomyosin contractility. (B). n = 357 (MDCK), 351 (+Caly), 434 (DMSO); N = 3. Kruskal-Wallis Test, followed by Dunn's multiple comparison Test; ***, P<0.001; ns, P>0.05. Desmosome width in MDCK cells becomes significantly lower upon reduced actomyosin contractility.

The reviewer's comments also motivated us to examine if tension-dependent DP conformational changes occur in non-epithelial systems. We therefore performed new experiments to examine whether DP undergoes similar conformational changes in primary cardiomyocytes (CMs) obtained from neonatal mice. In these mechanically active cells, desmosomes are positioned either perpendicular (axial) or parallel (lateral) to contractile myofibrils (Fig. R4A & R4C). Using STED microscopy, we compared Dsg2/DP/F-actin-stained CMs under contractile (control) and non-contractile (Blebb-treated) conditions and measured the width of axially and laterally oriented desmosomes. In axially aligned junctions, Blebb significantly reduced desmosome width (227 ± 62 nm in control vs. 178 ± 47 nm in Blebb; Fig. R4B). In contrast, laterally aligned junctions showed no significant difference between the control and +Blebb condition (172 ± 44 nm vs. 171 ± 47 nm; Fig. R4D). These results suggest that axially oriented desmosomes, like those in MCF7 WT cells, experience greater mechanical tension and adopt

Figure R4: Desmoplakin undergoes force-dependent conformational changes in primary cardiomyocytes. (A) Representative STED images of CMs from neonatal mice treated with DMSO (CM, **a**) or Blebbistatin (CM+Blebb, **b**), immunolabeled for DPC (red), Dsg2 (yellow), and F-actin (cyan). Overlays show proper desmosome formation at cell-cell contacts, axially aligned with contractile myofibrils. Scale bar: 3 μm. Insets display characteristic DP "railroad track" patterns at the junctional membrane. (B) Quantification of axially aligned desmosome widths (DPC-DPC distance) in CM and CM+Blebb cells. n = 384 (CM), 330 (CM+Blebb); N = 3. Mann-Whitney's U test; ***, P<0.001. (C) Representative STED images of CM (**a**) and CM+Blebb (**b**) showing desmosomes laterally aligned relative to the contractile axis. (D) Quantification of laterally aligned desmosome widths (DPC-DPC distance) in CM and CM+Blebb cells. n = 239 (CM), 297 (CM+Blebb); N = 3. Mann-Whitney's U test; ns, P>0.05.

an elongated morphology. In contrast, laterally oriented desmosomes, similar to those in K19-KO cells, are under lower tension and remain compact. Taken together, these findings support the conclusion that tension-dependent DP elongation is a conserved mechanism in both epithelial and mechanically active non-epithelial cells. We present these new findings in **lines 348-413** of the Result section and include a new **Figure 7**.

I also wonder if there could be non-specific effects of calyculin A in these responses, given its broad inhibition of phosphatases. Although it does lead to activation of myosin, it is also conceivable that it increases phosphorylation of desmosomal or keratin-associated proteins to change the conformation of DSP.

Response 12: We thank the reviewer for raising this important point. Since Calyculin A has been widely used to enhance actomyosin contractility in cultured cells [2, 3], we used Calyculin A as a well-established tool to examine how increased actomyosin force impacts desmosome morphology. However, we acknowledge that Calyculin A may also affect other phosphorylation-dependent pathways, including those involved in cytoskeletal regulation or desmosomal protein function, and have now included this limitation in the discussion section (**lines 472-475**).

4. Can the authors include representative images in Figure 4 for the calyculin A treatments, similar to those shown in 4C, D, H, and I? To

Response 13: To address this comment, we have included representative images of K8 filament organization for both the Blebbistatin and Calyculin A-treated WT and KO cells (Fig. R5 and R6 below). This result is now displayed in **Supplementary Figures S12 & S13**.

Figure R5. Keratin filaments reorient in response to reduced actomyosin forces with Blebbistatin treatment. (A-B) Representative confocal images of WT+Blebb (A) and KO+Blebb (B) cells immunolabeled for K8 (green) and DP (red). Scale bar: 5 µm (left panel). A magnified grayscale image highlights DP-associated K8 filaments at the junction with enhanced contrast (middle panel). The local dominant orientation of K8 filaments is represented as a vector field image (right panel).

Figure R6. Keratin filaments reorient in response to increased actomyosin forces with Calyculin A treatment. (A-B) Representative confocal images of WT+Caly (A) and KO+Caly (B) cells immunolabeled for K8 (green) and DP (red). Scale bar: 5 µm (left panel). A magnified grayscale image highlights DP-associated K8 filaments at the junction with enhanced contrast (middle panel). The local dominant orientation of K8 filaments is represented as a vector field image (right panel).

Reviewer #3 (Remarks to the Author)

The manuscript by Dong et al. uses super-resolution imaging, computer simulations and a novel FRET-based tension sensor for desmoplakin to show that actomyosin-mediated forces contribute to conformational changes in desmoplakin. The authors suggest that forces induce a mechanosensitive conformational change in the N-terminal plakin domain of desmoplakin, changing it from a folded to an extended conformation.

The manuscript is very well written and the experiments are well presented and analysed, but essential experiments to provide evidence for the claims made are missing. In addition, while it is interesting that desmoplakin appears to be under force, there is very little additional mechanistic and molecular insight into the functional consequences of the hypothesised conformational changes.

We thank the reviewer for their suggestions. We have now performed additional experiments and analysis to address the reviewer's comments.

Main weaknesses:

1) The authors suggest that a mechanosensitive conformational change in the N-terminal plakin domain of desmoplakin is responsible for the widening of tramlines in desmosomes that occurs under tension. While this may be plausible and the section looks interesting as the authors have identified some key residues that may be involved in opening this hinge, this model of extended conformation needs to be experimentally validated. It would be essential to show that removing the hinge region or introducing mutations that inhibit the conformational switch abolishes the widening of the tramline.

Response 14: We thank the reviewer for this suggestion. As described in Response 6, experimentally restricting DP conformational changes via targeted mutagenesis or truncation constructs would indeed provide important mechanistic insight. However, these are immensely challenging experiments given the substantial size of the DP protein and given that crystal structures of only a small fragment of the protein have been solved. Furthermore, as described in Response 3, the structural insights we obtain from AlphaFold predictions and MD simulations are inherently limited. We therefore believe that the type of structure-function analysis proposed by the reviewer is beyond the scope of our current study. We view the reviewer's suggestion as a critical direction for future work and have emphasized the importance of specific domains or residues in DP conformational regulation in **lines 538-545** of the Discussion section.

To address the reviewer's broader concerns regarding the absence of adequate mechanistic insights, we now perform additional experiments that are described in Response 8. These new experiments provide additional mechanistic insights into how K19 regulates mechanical force on desmosomes. We show that K19 plays a key role in organizing F-actin filaments at desmosomes and regulating the level of actomyosin forces integrated into the desmosomal complex.

2) Why did the authors use calyculin A to induce actin stress fibres? This can have so many different side effects. Activating RhoA is the direct way to induce actomyosin activation. However, the effect on force generation needs to be shown in a direct way. In addition, the actin close to the desmosomes, even at high resolution as shown in Figure 2b, is not very convincing because it is not clear whether they would interact and whether forces are exerted at these sites. In fact, it is very unlikely that they do so at these sites, since the cortical actin ring would rather be determined by adherens junctions in the vicinity. Also, in this case, the keratins are not oriented perpendicular to the desmosomes and therefore represent desmosomes under force.

Response 15: As noted in Response 12, since Calyculin A has been widely used to enhance actomyosin contractility in cultured cells [2, 3], we used Calyculin A as a well-established tool to examine how increased actomyosin force impacts desmosome morphology. However, we acknowledge that Calyculin A may also affect other phosphorylation-dependent pathways, including those involved in cytoskeletal regulation or desmosomal protein function. We therefore interpret the results with caution and have included Calyculin A's limitation in the discussion section (**lines 472-475**).

We agree with the reviewer that Figure 2B alone does not provide definitive evidence of force transmission or direct interaction at desmosomal sites. Our intent with this figure was not to imply functional force coupling, but rather to illustrate the spatial proximity between K19, DP, and F-actin, supporting a model in which these components may interact. To further validate this possibility, we performed co-IP experiments, which confirmed that K19 can associate with both DP and F-actin, as shown in Figure 2C. These data provide complementary biochemical evidence for a potential linkage between the desmosomal components and actin network via keratin cytoskeleton, which may contribute to force integration. In the revised manuscript, we have modified the language to interpret this point more cautiously (**lines 130-136**).

Finally, as described in Response 11, we reinforce our findings that actomyosin forces induce a conformational change in DP, by performing STED imaging in primary cardiomyocytes. These data demonstrate that in primary cardiomyocytes, DP undergoes an actomyosin force-dependent conformational change. Like MCF7 cells, where DP conformational change depended on actin filament orientation, we observe that DP conformational change in cardiomyocytes depends on myofibril orientation.

3) The western blots are not convincing. (i) they consist of excised regions and do not show the whole blot with appropriate MW indications and (ii) both keratin and actin are so abundant in cells with multiple indirect cross-talk possibilities. To show a direct desmosomal link, the authors should perform desmoplakin IPs in cells with and without K19 and with desmoplakin mutants that cannot bind to intermediate filaments (e.g. with those that were used in tension sensor experimnts).

Response 16: We thank the reviewer for their comment.

(i) As with all Nature Communications submissions, the complete, uncropped Western Blots for all replicates of all images are included in the **source data file**. The cropped western blot regions were intended to highlight specific differences in immunoblot (IB) signal between the IgG control and the K19 pull-down groups, supporting potential interactions between K19 and

desmosomal or cytoskeletal proteins. We have now added molecular weight (MW) markers to the cropped western blots.

(ii) To further address this concern, we performed new co-IP experiments using DP pull-down in both MCF7 WT and K19-KO cells. The results, as illustrated in Fig. R7 below, showed that DP interacts with keratin and actin filaments in both cell types, suggesting direct associations between DP and these cytoskeletal networks. This new data is now presented in **lines 146-150** and shown in **Supplementary Figure 5**. Unfortunately, we could not perform these experiments with FRET DP mutants that cannot bind to intermediate filaments due to the low transient transfection efficiencies of the cells.

Figure R7. Co-IP result for DP shows an interaction between DP and keratin and F-actin filaments. Co-IP of DP in WT and K19-KO cells. Co-IP was performed using anti-DP antibody, or an IgG control. The Co-IPs reveal that DP interacts with K8, K18, K19 and F-actin filaments in WT cells and K8, K18, and F-actin filaments in K19-KO cells.

4) Just showing a tension sensor and possible conformational changes in a protein is rather sparse information. The authors state in their discussion that "it is likely that force-induced opening of DP also exposes cryptic binding sites for junctional proteins that signal and remodel the desmosome and intermediate filaments". Without showing this, or exactly how actin contributes to the hinge switch, or whether perturbing the switch has functional consequences, there is little conceptual advance in the field.

Response 17: We thank the reviewer for raising this important point regarding the broader significance of our findings. We agree that while our current study establishes a correlation between mechanical tension and conformational changes in DP, the downstream consequences of these structural transitions — such as modulation of protein interactions or cell signaling— remain to be elucidated.

To acknowledge this limitation, we have revised the Discussion in **lines 527-537** to clarify that our proposed model is speculative and that further studies are needed to elucidate the mechanistic details of force-induced DP conformational changes and their impact on downstream signaling. As noted in Responses 6 and 7, strategies such as mutational approaches to stabilize either the folded or unfolded states of DP and proximity-based interactome mapping are promising avenues for future investigation, although they are beyond the scope of the current study. Nevertheless, we believe our findings offer a conceptual framework and foundational evidence that will guide future mechanistic studies of force-regulated desmosomal remodeling and its biological significance.

5) The most obvious control experiment for force contribution and tension sensors, i.e. blebbistatin treatment to inhibit actomyosin, has not been done. If these tension sensors work, then there is the opportunity of using them in living cells ± induction & inhibition of forces. The strength of intramolecular FRET probes and authors should be able to show the transition from high to low force and vice versa including changes in FRET changes in living cells.

Response 18: We thank the reviewer for raising this important point. Motivated by the reviewer's comment, we performed new control experiments to measure the FRET-based tension at DP in WT and K19-KO cells treated with Blebbistatin. As shown in Fig. R8, in WT cells treated with Blebbistatin (WT+Blebb), the FRET efficiency of DP-TS increased markedly to a level comparable to the control (Fig. 6F). This suggests a significant reduction in DP tension due to decreased actomyosin contractility. In contrast, K19-KO cells treated with Blebbistatin (KO+Blebb) showed no significant change in FRET efficiency between the TS and the control (Fig. 6G), indicating that DP in K19-KO cells already experiences minimal mechanical tension. These findings further support our model in which actomyosin-generated forces are transmitted to desmosomes in a KIF-dependent manner, influencing the mechanical tension experienced by DP. This new data is presented in **lines 336-346**, and shown in **Fig. 6F-G**. Unfortunately, due to the low transient transfection efficiencies of the FRET tension sensors, we were unable to perform live-cell FRET experiments.

Figure R8. Tension at DP is reduced as a result of lower actomyosin contractility in WT cells. (A) Quantification of FRET efficiency for DP-TS and no-tension control constructs in WT+Blebb cells. The median FRET efficiency of the tension sensor increased to a similar level to the control, indicating that Blebbistatin treatment significantly reduced the tension at desmosomes in WT cells. $n = 30$ images, $N = 3$, $P < 0.05$. **(B)** Quantification of FRET efficiency for DP-TS and no-tension control constructs in KO+Blebb cells. The median FRET efficiency of DP-TS was indistinguishable from that of the control, suggesting that DP experiences undetectable tension in KO+Blebb cells. $n = 30$ images, $N = 3$, $P > 0.05$.

6) If cells still have K8/18, why does DP remain in a closed conformation in K19 KO cells (line 300)? The authors speculate that the loss of K19 alters the organisation of keratin filaments to become more cortical and transmit less force through DP, but their keratin orientation data is not strong enough to support this at present. It would be interesting to see the effect of K18 KO for comparison and to test the hypothesis that K19 orients filaments radially in WT cells.

Response 19: As detailed in Response 8, our new data (**lines 184-212 and Figure 3**) support a key role for K19 in organizing F-actin filaments and anchoring them to desmosomes, thereby facilitating the transmission of actomyosin-generated forces to adhesive sites. In addition to keratin organization, we show that DP conformational changes are also influenced by actin enrichment at junctions. The reduced junctional F-actin in K19-KO cells likely results in diminished mechanical tension at desmosomes, weakening force integration at cell-cell contacts. While our findings highlight the importance of K19 in this process, we acknowledge that the specific contributions of other keratins, such as K18, remain to be clarified. Investigating the effects of K18 depletion would be a valuable future direction, though it is beyond the scope of the current study.

7) There is very little mechanistic insight into the contribution of the different keratins to forces. All keratins appear to interact with actin, but only a subset are efficient force transducers. As there is little mechanistic insight, this section remains purely descriptive and raises many questions rather than providing functional insight. The apparent differences seen in MDCK and MCF7 cells are also difficult to understand and raise additional questions.

Response 20: We appreciate the reviewer's important point regarding the limited mechanistic insight into the differential roles of various keratins in force transmission. In this study, we focused specifically on K19, which we found to be essential for linking actomyosin-generated forces to desmosomes via its role in organizing F-actin filaments and coupling them with DP, as mentioned in Response 19. While it is beyond the scope of the current study, we agree that a more comprehensive dissection of how individual keratin isoforms differ in their ability to organize actin remains to be fully explored.

Regarding the observed differences between MDCK and MCF7 cells, as outlined in Response 11, we have included new data demonstrating that desmosomes in MDCK cells become significantly more extended under high-tension conditions induced by Calyculin A, and more compact under reduced tension with Blebbistatin treatment. Similarly, as described in Response 11, this interpretation is significantly enhanced by our data showing that the DP in primary CMs becomes compact when the CMs are treated with Blebbistatin. Taken together, these findings now demonstrate that tension-dependent DP elongation is a conserved mechanism in both epithelial and mechanically active non-epithelial cells and is not limited to MCF7 cells.

Minor points:

- The authors' description of the desmoplakin structure (lines 67-68) seems to me to be a bit off. I believe that the globular head region generally encompasses the entire N-terminus (everything before the rod domain) and includes spectrin repeats 1 and 2 as well as the downstream plakin

domain. However, the authors seem to refer to the region before the plakin domain as the globular head domain.

Response 21: We apologize for the confusion, which stems from context-dependent differences in terminology. In structural biology, the term *globular head* typically refers to the entire N-terminal region of desmoplakin, as the reviewer correctly notes. However, in the context of domain-mapping studies such as ours, the *globular head* is used to describe only the extreme N-terminal ~200 residues, located upstream of the plakin domain. To avoid ambiguity, we have revised the manuscript in **lines 70-71** to clarify our usage of this term.

- As a general observation, the authors seem to use subconfluent monolayers of MCF-7 cells throughout the study. I wonder if this is suboptimal for establishing uniform tension across the epithelial sheet. On the other hand, polarised migrating cells, which are likely to exert more force on desmosomes than cells in confluent areas, could be more rigorously analysed for their cytoskeletal appearance (F-actin, keratin and desmosomes), which should change under blebbistatin treatment.

Response 22: We thank the reviewer for this important observation. Indeed, we performed most of our experiments using subconfluent monolayers of MCF7 cells because K19-KO cells tend to cluster and grow in multiple layers when cultured at higher densities, making it difficult to maintain a uniform monolayer for imaging and analysis. To ensure valid comparisons, we carefully controlled cell density and seeded WT and K19-KO cells at equivalent densities for all experiments. While we acknowledge that higher confluency may help establish more uniform epithelial tension, our approach was optimized to balance monolayer integrity with the ability to perform meaningful comparisons across genotypes. We have clarified this point in the Methods section in **lines 560-562**, noting that subconfluent monolayers were used to prevent multilayering in K19-KO cells and to ensure comparable conditions between WT and K19-KO lines

- Figure 4A is oddly cropped in the 'no tension' schematic. The comparison between K8 filaments in WT and K19-KO cells (line 220) is not convincing because the representative images chosen are not comparable. The junctions shown in wild-type cells are shorter, as if formed between two lamellipodia, whereas the K19-KO image shows a long continuous junction between two adjacent cells. This also applies to the K18 images in Figures 4H and should be corrected when published. The authors also fail to show representative images of drug-treated samples. These should at least be included in the Supplements.

Response 23: We respectfully do not understand why the reviewer feels that the no-tension schematic for Fig. 4A in the original submission (Fig. 5A in the new submission) is 'oddly cropped'. We have left the figure unchanged since we did not know what the reviewer would like us to change.

To improve clarity and enable direct comparison between the cell lines, we have selected more representative images of K8 and K18 filaments in WT and K19-KO cells. These revised images show long, continuous junctions between adjacent cells, facilitating a clearer assessment of filament organization. The updated images are now included in Fig. R9 below. Additionally, as

shown in Response 13, we have included representative images of K8 filament organization for the Blebbistatin and Calyculin A-treated WT and K19-KO cells, which are illustrated in the Fig. R5-6. This result is now displayed in **Supplementary Figures S12 & S13**.

Figure R9. Revised representative images for K8 and K18 filaments in WT and K19-KO cells. Revised representative images for K8 filaments in WT and K19-KO cells (left panel) and revised representative images for K18 filaments in WT and K19-KO cells (right panel).

- The authors use dispase (line 341) to lift sheets of WT or K19-KO MCF-7 cells and incubate them in EGTA for 1 hour to select for hyperadhesive desmosomes. To my knowledge desmosomes reach only hyperadhesion during prolonged cell culture? Under the conditions in which the cells are cultured in this manuscript, it is likely that they are still Ca-dependent. The question is whether EGTA has a limited effect on cell-cell adhesion under these conditions and the disruption is due to increased fragility of the intermediate filaments?

Response 24: We agree that desmosomes typically require prolonged cell-cell contact and specific culture conditions to reach a hyperadhesive, Ca²⁺-independent state. In our study, we did not claim, or intend to imply, that the desmosomes were in a hyperadhesive state. Rather, the dispase-based mechanical dissociation assay was used solely to compare relative adhesion strength between WT and K19-KO MCF7 cells under identical conditions. We treated the cells with EGTA merely to eliminate the contribution of classical cadherins to cell-cell adhesion. We however acknowledge that EGTA would also partially disrupt desmosomes, although this process would likely occur equally in the WT and K19-KO cells. To clarify this point, we have revised the manuscript text in **lines 500-507** to avoid the implication of hyperadhesion and to emphasize that our interpretation is based on comparative differences in mechanical resistance.

Reviewer #4 (Remarks to the Author)

We thank the reviewer for their suggestions. We have now performed additional experiments and analysis to address all the reviewer's comments.

References

1. Kurochkina, N. and U. Guha, *SH3 domains: modules of protein–protein interactions*. Biophysical Reviews, 2013. **5**(1): p. 29-39.
2. Nanavati, B.N., et al., *The desmosome-intermediate filament system facilitates mechanotransduction at adherens junctions for epithelial homeostasis*. Current Biology, 2024. **34**(17): p. 4081-4090. e5.
3. Wyatt, T.P., et al., *Actomyosin controls planarity and folding of epithelia in response to compression*. Nature materials, 2020. **19**(1): p. 109-117.

Response to Reviewers

We appreciate all the reviewers' comments and suggestions. Below is our point-by-point response to the reviewers. The original reviewer comments are highlighted in *blue*, and our response is in black.

Reviewer #1 (Remarks to the Author):

The authors have thoroughly addressed all of my comments, concerns, and suggestions, and have revised the manuscript accordingly. I support the publication of this work.

We thank Reviewer #1 for their positive feedback.

Reviewer #2 (Remarks to the Author):

In this revised manuscript the authors have made substantial improvements to the manuscript. In particular, there is new evidence showing that DSP changes in length in cardiomyocytes, as well as when contractility is increased in MDCK cells.

This reviewer remains very excited by the new finding that DSP may experience mechanosensitivity through a possible hinge/unfolding region, and that this finding appears to occur in multiple cell types (the addition of the cardiomyocytes further increases the significance/importance of their findings). The authors data establishes that the length is changing dependent on actomyosin contractility.

While I am very curious how this 'hinge' region works and what are the biological consequences of it, I do feel that it would be a disservice to hold back this publication for some time while the

authors figured this out. Thus, I believe the scientific community is better served by having these findings published vs waiting for additional information and context.

We thank Reviewer #2 for their supportive comments and positive feedback.

Reviewer #3 (Remarks to the Author):

Many of the data have improved and are of high quality. However, it is rather unfortunate that the authors have not addressed, in my opinion, the most important question, i.e. the modification of the hinge region to show that the hypothesised conformational change is true. While DP is a large protein, the modern cloning strategies and transfection possibilities should enable these deletions/modifications and expression in cells. While it may be challenging, I don't think it is "immensely challenging" but rather doable. It seems that the authors have done cloning and transfections sufficiently to analyse the existing tension sensors and it is difficult to understand why they cannot do modifications and transfections to verify their hypothesis.

Similarly, the AlphaFold predictions are quite powerful and can be used to design mutations to bolster or disrupt predicted residue interactions in the closed conformation. Again this would be easy to do, particularly on the basis that essential parts of the manuscript are based on modelling and MD simulations. Additionally, it would be nice to see the AlphaFold confidences projected onto their model which would aid in identifying confidently predicted interactions.

I feel that this is not beyond the scope of this manuscript; it is the focal point of the manuscript which should be verified and it provides the real novelty for publication in Nat Comms. Novelty without these data are marginal since DP tension sensors have been published previously showing that it is under mechanical load under specific conditions.

We thank the reviewers for their suggestions. We have addressed the reviewer's minor comments below.

Minor points:

The inclusion of FRET control experiments in the presence of blebbistatin adds weight to the actomyosin mediated force hypothesis. In line with their other experiments, it would have also been useful to see whether the FRET readout implies increased DP tension upon calyculin A treatment.

Response 1: We appreciate the reviewer's suggestion. While we attempted to perform the FRET experiments that the reviewer suggested, we faced a technical limitation. The addition of calyculin A, which increases cell contractility, resulted in the rupture of many cell-cell junctions. Given the already low transfection efficiency of our large DPI-TS constructs, we were unable to obtain sufficient, quantitative FRET data. However, we believe that our blebbistatin experiments already provide direct support for the role of actomyosin forces and are consistent with the other experimental evidence presented in the manuscript. We now describe the technical limitations of the calyculin A experiments on **lines 343-345**.

Response 24 still doesn't justify, in my opinion, why the authors chose to use EGTA at all. Especially for an extended period, one would expect to lose all desmosome mediated adhesion? As far as I am aware, it is not a normal step in the dispass assay.

Response 2: We thank the reviewer for seeking this clarification. Our decision to include EGTA was to minimize adhesion mediated by classical cadherins, to better assess the relative

contribution of desmosomes to intercellular cohesion in WT vs. K19-KO cells. We would like to point out that use of EGTA in dispase assays is not uncommon (see references [1, 2]). While we recognize that EGTA also weakens desmosomal adhesion, this weakening applies equally to both WT and K19-KO cells. Importantly, we consistently observed reproducible differences between WT and K19-KO cells, which supports our conclusion that keratin-19 knockout influences desmosome-mediated adhesion. Our manuscript explicitly emphasizes that our interpretation is based on comparative differences rather than absolute adhesion strength. On **lines 505-506**, we now cite previous publications that used EGTA in a dispase assay.

1. Garrod, D.R., et al., *Hyper-adhesion in desmosomes: its regulation in wound healing and possible relationship to cadherin crystal structure*. Journal of cell science, 2005. **118**(24): p. 5743-5754.
2. Wallis, S., et al., *The α isoform of protein kinase C is involved in signaling the response of desmosomes to wounding in cultured epithelial cells*. Molecular biology of the cell, 2000. **11**(3): p. 1077-1092.

Reviewer #4 (Remarks to the Author):

We thank Reviewer #4 for co-reviewing our manuscript.